# MGP: a new 1-hourly 0.25° global precipitation product (2000-2020) based on multi-source precipitation data fusion

Hanqing Chen[a], Debao Wen[a*], Bin Yong[b*], Jonathan J. Gourley[c], Leyang Wang[d], Yang Hong[e]

*[a]School of Geography and Remote Sensing, Guangzhou University, Guangzhou 510006, China.*

*[b]State Key Laboratory of Hydrology-Water Resources and Hydraulic Engineering, Hohai University, Nanjing 210098, China.*

*[c]National Oceanic and Atmospheric Administration (NOAA)/National Severe Storms*

*Laboratory, Norman, OK 73072 USA.*

*[d]Faculty of Geomatics, East China University of Technology, Nanchang 330013, China.*

*[e]Clayton School of Information Technology, Monash University, Melbourne 3000, Australia.*

*\*Corresponding author: Prof. Debao Wen and Prof. Bin Yong*

*Email: wdbwhigg@gzhu.edu.cn and yongbin@hhu.edu.cn*

*Affiliation: School of Geography and Remote Sensing, Guangzhou University,*

*Guangzhou 510006, China (Debao Wen) and State Key Laboratory of Hydrology-Water Resources and Hydraulic Engineering, Hohai University, Nanjing 210098, China (Bin Yong).*

**Abstract:** a high-quality global precipitation product with finer spatiotemporal
resolutions and long-term temporal coverage is critical for a variety of science

communities (e.g., hydrology, meteorology, climatology, ecology, and agriculture).
Here, a novel multi-source precipitation data fusion (MPDF) algorithm, which
considers the dependency of precipitation errors on seasonality, was proposed to fully
take advantage of the complementary strengths from satellite, reanalysis, and gauge
data for generating a higher-quality global precipitation product. Two merging schemes,

which used six products (including four satellite precipitation products: IMERG-Late,
GSMaP-MVK, TMPA-RT, and PERSIANN-CCS; one reanalysis precipitation product
ERA5; one ground-based precipitation product CPCU) and three products (i.e.,
IMERG-Late, ERA5, and CPCU) as input data sources of the MPDF algorithm
respectively, were designed to generate two different high-quality multi-source merged

global precipitation products (MGP), i.e., MGP-6P and MGP-3P. The results show that
the proposed MPDF algorithm is effective in considering the advantages from satellite,
reanalysis, and gauge data. Global comparisons indicate that the MGP suite products
with regard to daily mean precipitation share a similar spatial pattern with other global
precipitation products (i.e., MSWEP, IMERG-Final, GSMaP-Gauge, ERA5, and CPCU)

in most overland regions globally; while large differences between these seven products
occur in Australia, southeast China, Europe, near the equator of Africa and South
America, and so on. Overall, MGP-3P substantially performs better than the other five
research-quality products (i.e., MGP-6P, MSWEP, IMERG-Final, GSMaP-Gauge, and
ERA5) in the ground validation on the Chinese mainland, with the highest POD, CC

and lowest RMSE of 0.85, 0.71, and 1.21 mm, respectively, at a 3 hourly scale.

Especially, the accuracy and detection capability of MGP-3P are the best in most hourly

rainfall intensity groups. The MGP-3P product can provide a new precipitation data

option for research and applications in the field of hydrology, meteorology, climatology,

ecology, and agriculture. MGP-3P (also known as MGP) Version 1.1 is available at the

following link: https://www.zenodo.org/record/7386441#.Y8zr4clBxD9 (Chen et al.,

2022a).

**Keywords:** MGP; precipitation; multi-source precipitation data fusion; performance

evaluation; global precipitation product



## 1. Introduction

Precipitation is one of the main components of the global water cycle and global energy cycle, as well as the main input for a variety of Earth system models (Daly et al., 1994; Trenberth et al., 2007; Skofronick-Jackson et al., 2017). Meanwhile, accurate precipitation information is critical for various operational applications such as climate change analysis, hydrological simulation, water resources management, and the monitoring and forecasting of typhoons, flood landslides, and other precipitation-related disaster events (Hou et al., 2014; Maggioni et al., 2016; Sun et al., 2018; Ma et al., 2020; Chen et al., 2022b). However, precipitation is one of the most difficult meteorological variables to measure due to its high spatiotemporal heterogeneity (Adler et al., 2001; Stephens et al., 2010; Beck et al., 2017).

Currently, precipitation amount estimates are mainly from rain gauges, weather radars, atmospheric retrospective-analysis models, and satellite precipitation retrievals (Kidd and Huffman, 2011; Maggioni et al., 2016; Kidd et al., 2017; Sun et al., 2018). Among them, precipitation measurements from ground-based rain gauges and weather radars are the most reliable source of precipitation (Hijmans et al., 2005; Herold et al., 2015; Kidd et al., 2017). Therefore, ground-based precipitation observations have normally been regarded as a ground reference for validating and correcting satellite and reanalysis precipitation estimates (Tian et al., 2010; AghaKouchak et al., 2012; Chen et al., 2022b; Sun et al., 2022). Nevertheless, the quality of gauge observations is extremely dependent on the spatial density of the rain gauges (Krajewski et al., 2000; Villarini and Krajewski, 2007; Roca et al. 2010; Prakash et al., 2019; Chen et al., 2020a).

In addition, gauge observations are subject to systematic biases to some extent, primarily due to wind-induced under-catch (Rasmussen et al., 2012; Kidd et al., 2017). More importantly, ground-based global gridded precipitation products (see

supplementary material Table S1) have considerable uncertainties in most regions where there is a spare and uneven spatial distribution of the rain gauges, especially for ungauged areas (Rudolf et al. 1994; Xie and Arkin, 1997; McCollum and Krajewski, 1998; Villarini et al., 2008; Ouyang et al., 2021). An interpolation-based error would be introduced in ground-based global gridded precipitation products during the

interpolating procedure (Xie et al., 2007; Chen et al., 2008; Xie and Xiong, 2011; Nie et al., 2015; Shen et al., 2016). These objective issues existing in ground-based global gridded precipitation products make them difficult to meet the requirements of various operational applications. In terms of weather radars, they are relatively expensive to purchase, operate, and maintain, resulting in limiting their availability in many regions

of the world, especially in developing countries and the ocean (Dinku et al., 2002; Yang et al., 2006; Zhang et al., 2016; Chen et al., 2019). Also, precipitation estimates from ground-based weather radars are subject to random and systematic errors, because this precipitation retrieval technique is impacted by the environment and the instrument itself (Zhang et al., 2016; Chen et al., 2019). Up to now, we still cannot comprehensively

obtain reliable global precipitation estimates by depending on currently deployed networks of weather radars and rain gauges (Kidd et al., 2017).

Satellite precipitation retrievals provide an effective way for hydrometeorologists to obtain instantaneous and continuous precipitation information for large domain areas

(Hou et al., 2014; Maggioni et al., 2016). Currently, multiple (quasi)global satellite

precipitation products have been published by several precipitation teams.

Supplementary material Table S1 provides an overview of the currently popular

(quasi)global satellite precipitation products. Global satellite precipitation products are

a good supplement to ground-based global precipitation products, with finer

spatiotemporal resolutions, spatial continuity, and near real-time acquisition (Kidd and

Huffman, 2011; Gehne et al., 2016; Sun et al., 2018). Especially, satellite precipitation

products have provided unprecedented abundant precipitation estimates over remote

mountain areas, the ocean, and other ungauged areas, which fills the gap left by ground

precipitation observations. Meanwhile, satellite precipitation retrievals are good at

measuring the precipitation in the tropics, especially for the areas near the equator,

because both passive microwave (PMW) onboard low-Earth orbit satellites and infrared

(IR) onboard geostationary satellites are good at observing convective precipitation

occurrences and their magnitude of precipitation (Smith et al., 2005; Dinku et al., 2008;

Chen et al., 2020b). Yet, satellite precipitation retrievals still have several limitations:

(1) satellite sensors are insensitive to light and solid precipitation, although a dual-

frequency precipitation radar (DPR) and a multichannel global precipitation

measurement (GPM) Microwave Imager (GMI) are used to extend the capability of

satellite precipitation retrievals for further improving the measurement of light and solid

precipitation (Hou et al., 2014); (2) satellite precipitation retrievals may not capture the

precipitation from the warm clouds as cloud-top of the warm clouds would be too warm

for IR thresholds and there is not enough ice loft to be detected by PMW sensors (Dinku

et al., 2008); (3) satellite sensors might fail to identify precipitation occurrences when surface lands are covered with snow and ice (Ferraro et al., 1998; Kidd and Levizzani, 2011); (4) the measurement of orographic rains is still one of the difficulties that need to be solved in satellite precipitation retrievals (Xu et al., 2017; Hashemi et al., 2017;

Chen et al., 2019). As for atmospheric reanalysis models, they do well in simulating the evolution of large-scale weather systems, whereas they are difficult to show convection-related variability and have limitations in the parameterizations of sub-grid processes (Roads, 2003; Kidd et a., 2013; Beck et al., 2017).

A large number of papers executed the evaluation of global satellite and reanalysis

precipitation products on global and regional scales and found that global satellite and reanalysis precipitation products are subject to large errors and uncertainties (AghaKouchak et al., 2011; Yong et al., 2015; Takido et al., 2016; Tan et al., 2016; Gebregiorgis et al., 2018; Xu et al., 2022; Chen et al., 2020b, 2021, 2023). To this end, many studies proposed various error adjustment methods and multi-source precipitation

data fusion algorithms to reduce errors existing in global satellite and reanalysis precipitation products. Several papers established statistical relationships between satellite estimates and ground observations to reduce errors in global satellite precipitation products (e.g., Xiong et al., 2008; Tian et al., 2010; Shen et al., 2014; Yang et al., 2016); satellite soil moisture information was unitized to adjust the satellite

precipitation estimates (e.g., Crow et al., 2011; Wanders et al., 2015; Massari et al., 2019); some studies established error adjustment models through considering some crucial impacting factors (i.e., topography, seasonality, climate type, and rain rate; e.g.,



Hashemi et al., 2017; Chen et al., 2022b); spatial interpolation was proved to be an effective method in improving the quality of global satellite and reanalysis precipitation

estimates (e.g., Li and Shao, 2010; Xie and Xiong, 2011; Woldemeskel et al., 2013; Dinku et al., 2014; Nie et al., 2015); in recent years, machine learning was applied to enhance the quality of global satellite and reanalysis precipitation products (e.g., Ma et al., 2018a; Shen et al., 2019; Wu et al., 2020; Baez-Villanueva et al., 2020; Hong et al., 2021); also, weighted averaging of multi-source precipitation data has become a hotspot

in precipitation field, and many articles proposed a variety of multi-source data fusion algorithms, methods, or models to improve the quality of precipitation products (e.g., Beck et al., 2017, 2019; Wang et al., 2020; Lyu et al., 2020; Zhu et al., 2021). These algorithms, methods, and models have been proven to be effective in improving the quality of precipitation estimates and offer a potential alternative for data users to

improve the quality of precipitation estimates.

  However, most above-mentioned algorithms, methods, and models have only been validated in local areas, and their effectiveness on a global scale remains to be investigated. Currently, several popular gauge-adjusted global precipitation products (see supplementary material Table S1) have been generated by using various error

correction methods and multi-source data fusion algorithms, such as the Integrated Multi-satellitE Retrievals for Global Precipitation Measurement Final run (IMERG-Final; Huffman et al., 2019), the gauge-adjusted Global Satellite Mapping of Precipitation (GSMaP-Gauge; Mega et al., 2019), the Precipitation Estimation from Remotely Sensed Information using Artificial Neural Networks - Climate Data Record

(PERSIANN-CDR; Ashouri et al., 2015), the Climate Prediction Center (CPC)

MORPHing technique bias corrected (CMORPH-CRT; Joyce et al., 2004), the Climate

Hazards group Infrared Precipitation with Stations (CHIRPS; Funk et al., 2015), and so

on (see Supplementary material Table S1). The above gauge-adjusted global

precipitation products normally used ground-based precipitation observations as a

reference to correct corresponding satellite-only precipitation estimates, and hence this

way cannot consider the advantages from multiple types of precipitation information

sources (e.g., satellite, reanalysis, and ground precipitation observations). Beck et al.

(2017, 2019) proposed a Multi-Source Weighted-Ensemble Precipitation (MSWEP)

algorithm to consider the advantages of various precipitation data sources. Overall, the

MSWEP product performs better than most popular global precipitation products in

most areas of the world (Beck et al., 2017, 2019), but has a relatively coarse temporal

resolution of 3 hourly. In fact, the quality of the MSWEP product mainly depends on

that of the ground-based precipitation observations used in that MSWEP was derived

from temporal downscaling of high-quality ground-based precipitation observations

with a coarse temporal resolution, and relative weights designed to ground-based

precipitation observations are relatively larger than ones designed to satellite- and

reanalysis-based precipitation estimates (Beck et al., 2017, 2019). Besides, our previous

study found that MSWEP performs worse than two gauge-adjusted satellite

precipitation products (i.e., IMERG-Final and GSMaP-Gauge) in mainland China

(Chen, 2022c). More importantly, obtaining a reasonable weight is essential for the

quality of merged precipitation estimates because an optimal weight can be effective in

considering the advantages of various precipitation data sources. In summary, a new higher-quality global precipitation product with a fine spatiotemporal resolution, which is derived from a more advanced multi-source precipitation data fusion algorithm, is

vital for multifarious scientific research and operational applications.

Consequently, this study aims to propose a novel multi-source precipitation data fusion (MPDF) algorithm with considering the impact of seasonality on precipitation errors to develop a higher-quality global precipitation product with a 0.25° spatial and hourly temporal resolution for the period 2000-2020. Currently, a new global

precipitation product, namely multi-source merged global precipitation product (MGP) by the weighted merging of the three global precipitation products (MGP-3P), has been provided publicly to data users around the world.

## 2. Data and algorithm

### 2.1 Multi-source precipitation data fusion algorithm

Here, we proposed a novel multi-source precipitation data fusion (MPDF) algorithm, which considers the impact of seasonality on precipitation errors, to develop a higher-quality global precipitation product. Four satellite-only precipitation products (i.e., MERG-Late (Huffman et al., 2019), GSMaP-MVK (Ushio et al., 2009), TMPA-RT (Huffman et al., 2007), and PERSIANN-CCS (Hong et al., 2004)), one reanalysis

precipitation dataset (i.e., ERA5; C3S, 2017), and one ground-based precipitation product (i.e., CPCU; Xie et al., 2007; Chen et al., 2008), were utilized as the input data sources of the MPDF algorithm. Detailed information on the above global precipitation

products can be available in supplementary material Table S1. The flowchart of the
novel MPDF algorithm is displayed in Fig. 1, and the productive processes of MGP
mainly include five steps:

**Insert Fig. 1 about here**

**A.  Preprocessing of satellite and reanalysis precipitation data**

(1) All satellite and reanalysis data (except for TMPA-RT) were converted to 0.25°
spatial and hourly temporal resolutions.

(2) The time domain of all satellite and reanalysis data was firstly matched with that
of CPCU because of timing mismatches between satellite (reanalysis) and CPCU, and
then all satellite and reanalysis data were converted to 0.25° spatial and daily temporal
resolutions.

(3) The time domain of all satellite and reanalysis data was firstly matched with that
of CPCU, and then all satellite and reanalysis data were converted to 0.5° spatial and
daily temporal resolutions.

**B.  Obtaining global maps of the correlation coefficient of the satellites and
     reanalysis data**

(1) The precipitation amounts of all global precipitation products were separated
into four seasons (i.e., MAM (March-May), JJA (June-August), SON (September-
November), and DJF (December-February)) because of the dependency of precipitation
errors on seasonality. The correlation coefficient (CC) values of all satellite and
reanalysis data at the grid cells with at least one rain gauge for the four seasons (i.e.,
MAM, JJA, SON, and DJF) were calculated, using ground-based precipitation



observation data CPCU as a reference.

(2) Spatial maps of the CC of all satellites and reanalysis precipitation data for the
four seasons (i.e., MAM, JJA, SON, and DJF) were given by using the inverse-distance
weighting spatial interpolation (IDWSI) technique.

**C. Weighted merging of satellite, reanalysis, and ground precipitation**
**observations**

(1) Considering the CC score can fully reflect the performance of the satellite and
reanalysis products, the CC values were used as the weights for the satellite and
reanalysis products. As for ground-based CPCU, the quality of the ground precipitation
observations depends on the spatial density of the rain gauges (Krajewski et al., 2000;
Villarini and Krajewski, 2007; Roca et al. 2010; Prakash et al., 2019; Chen et al., 2020a);
therefore, the number of rain gauges at a grid cell was utilized as the weights for CPCU
at corresponding grid cell. The gauge precipitation observations used in the merging
procedure can be seen in Fig. 2a. The satellite and reanalysis precipitation data matched
with CPCU (0.5°, daily; i.e., (3) of subsection A) and the CPCU data were weighted
merging by the following eq. (1) to obtain merged global precipitation estimates,
namely product A (0.5°, daily).

$$P_m = \frac{P_{S_1} \times C_{S_1} + \cdots + P_{S_i} \times C_{S_i} + \cdots + P_{S_5} \times C_{S_5} + P_G \times n_G}{C_{S_1} + \cdots C_{S_i} + \cdots + C_{S_5} + n_G} \qquad (1)$$

where $P_m$ is merged precipitation estimates; $P_{S_i}$ is precipitation estimates from four
satellites (i.e., IMERG-Late, GSMaP-MVK, TMPA-RT, and PERSIANN-CCS) and
one reanalysis (i.e., ERA5), respectively, $i = 1, \cdots 5$; $C_{S_i}$ denotes CC values of
satellite and reanalysis data; $P_G$ is precipitation observations from CPCU; $n_G$ is the

number of rain gauges for each grid cell.

(2) The satellite and reanalysis precipitation data matched with the time domain of CPCU (0.25°, daily; i.e., (2) of subsection A) were weighted merging by eq. (1) to

calculate merged global precipitation estimates, namely product B (0.25°, daily). It should be noted that the weights of the satellite and reanalysis precipitation products at a 0.25° spatial resolution come from their CC values at a 0.5° spatial resolution. It should be noted that this step only merged satellite and reanalysis data, while CPCU data did not participate in the weighted merging procedure.

(3) all satellites (except TMPA-RT) and reanalysis precipitation products with 0.25° spatial and hourly temporal resolutions were weighted merging to compute merged global precipitation product, namely product C (0.25°, hourly). Similarly, the weights of the satellite and reanalysis precipitation products at a 0.25° spatial resolution come from their CC values at a 0.5° spatial resolution.

The merging processes in (1) of this subsection follow three criteria:

I. the precipitation estimates at the grid cells with at least one rain gauge come from the weighted merging of all satellite, reanalysis, and CPCU;

II. the precipitation estimates at the grid cells with no rain gauge observations come from the weighted merging of all satellite and reanalysis. In this case, the $n_G$ in eq. (1)

is set to zero. In other words, ground-based CPCU was abandoned in the weighted merging procedure because of possible existing large precipitation uncertainties at the grid cells with no rain gauges' observations;

III. the precipitation estimates come from CPCU precipitation observations when



the grid cells have gauge observations and rain gauges captured precipitation

occurrences but satellites and reanalysis did not.

**D. Spatial downscaling of the merged precipitation product A**

The weights of spatial downscaling were set to a ratio between two precipitation

estimates from different spatial resolutions, i.e., the precipitation value of a 0.25-degree

grid cell in the merged product B divided by the precipitation value of its corresponding

0.5-degree grid cell in the same product B. The equation is as follows:

$$W_{ij}^{S} = \frac{P_{ij}^{B_{0.25}}}{P_{kl}^{B_{0.5}}} \tag{2}$$

where subscript $ij$ indicates the grid cell located in row $i$ and column $j$; $P_{ij}^{B_{0.25}}$ is

precipitation value of 0.25-degree product B in the grid cell in row $i$ and column $j$; $P_{kl}^{B_{0.5}}$

denotes precipitation value of 0.5-degree product B in the grid cell in row $k$ and column

$l$.

A merged product, namely D (0.25°, daily), can be generated by multiplying ratio

$W_{ij}^{S}$ by merged product A. The specific calculation equation is as follows:

$$P_{ij}^{D} = W_{ij}^{S} \times P_{kl}^{A} = \frac{P_{ij}^{B_{0.25}}}{P_{kl}^{B_{0.5}}} \times P_{kl}^{A} \tag{3}$$

Noted that the precipitation values in four 0.25-degree grid cells (i.e., product D) were

set to the precipitation value in their corresponding 0.5-degree grid cell (i.e., product A)

when the merged product A captured precipitation occurrences but the merged product

B did not. At this stage, spatial downscaling is completed.

**E. Time downscaling of the merged precipitation product D**

The weights of time downscaling were set to a ratio between two precipitation

estimates from different temporal resolutions, i.e., an hourly precipitation value from

the merged product C divided by its corresponding daily precipitation value from the

merged product B. The equation is as follows:

$$W_i^t = \frac{P_i^C}{P_j^B} \tag{4}$$

where $i = 1,2,\cdots,24$; $j$ is corresponding day for $i$; $P_i^C$ indicates a precipitation value

of the $i^{th}$ hour in the j$^{th}$ day for product C; $P_j^B$ denotes a precipitation value of the j$^{th}$

day for product B. A merged product, namely E (0.25°, hourly), can be produced by

multiplying ratio $W_i^t$ by merged product D. The specific calculation equation is as

follows:

$$P_i^E = W_i^t \times P_j^D = \frac{P_i^C}{P_j^B} \times P_j^D \tag{5}$$

where $P_i^E$ denotes precipitation values of the $i^{th}$ hour in the $j^{th}$ day for product E; $P_j^D$

are precipitation values of the $j^{th}$ day for product D. In this step, MGP was generated

completely, i.e., the product E is MGP.

Merging multi-source precipitation estimates can realize the complementary

advantages of different precipitation data sources, so as to improve the quality of

precipitation (Beck et al., 2019; Ma et al., 2020). Nevertheless, each precipitation data

has its own limitations and deficiencies, and those limitations and deficiencies might

be propagated to merged precipitation estimates, resulting in a negative impact for the

quality of merged precipitation estimates. Therefore, this raises the question whether

the more precipitation data are merged, the better the merged precipitation estimates

will be. The solution to this question is conducive to providing constructive suggestions

for the subsequent research on the design of multi-source precipitation data fusion

algorithms. Here, we designed two merging schemes to generate two different MGP products, and a performance comparison between two different MGP products was used to solve this question. Two merging schemes were designed as follows:

**Scheme 1:** four satellite-only precipitation products (i.e., MERG-Late, GSMaP-MVK, TMPA-RT, and PERSIANN-CCS), one reanalysis precipitation dataset (i.e., ERA5), and one ground-based precipitation product (i.e., CPCU) were used as input sources of the MPDF algorithm to produce a new global precipitation dataset, namely MGP-6P.

**Scheme 2:** satellite-based IMERG-Late precipitation product, ERA5 reanalysis precipitation product, and ground-based CPCU precipitation product were used as input sources of the MPDF algorithm to generate another new global precipitation dataset, namely MGP-3P.

Many studies found that IMERG-Late performs better than other satellite-only

precipitation products (e.g., Tang et al., 2020; Chen et al., 2020b, 2021), and ERA5 has satisfactory detection capability and performs better in the winter season and high-latitude areas (Beck et al., 2017, 2019; Tang et al., 2020; Xu et al., 2022). Thus, satellite-based IMERG-Late, ERA5 reanalysis precipitation product, and ground-based CPCU were used as the input sources in scheme 2.

**2.2  Data and evaluation metrics**

The performance of the MPDF algorithm and MGP suite products (i.e., MGP-6P and MGP-3P) was evaluated at three time periods (i.e., daily, 3 hourly, and hourly). A

multi-source data merged product Version 2.8 MSWEP (hereafter refer to as MSWEP; Beck et al., 2017, 2019), two widely used and research-quality gauge-adjusted satellite

precipitation products Version 6 IMERG-Final (hereafter refer to as IMERG; Huffman et al., 2019) and Version 7 GSMaP-Gauge (hereafter refer to as GSMaP; Mega et al., 2019), and a state-of-the-art reanalysis precipitation product ERA5 (C3S, 2017) were used for comparison with MGP suite. MSWEP is a typical multi-source data fusion product with considering advantages from multiple precipitation data sources (Beck et

al., 2017, 2019); while both IMERG and GSMaP are gauge-adjusted global precipitation products, and have better performance than other global precipitation products in mainland China (Tang et al., 2020; Wei et al., 2021; Chen, 2022c); performance comparison between these three high-quality global precipitation products and the MGP suite products can strongly prove the effectiveness of the MPDF algorithm

and the excellence of MGP performance; in the end, selecting ERA5 as one of the products for comparison was to analyze potential error propagation in the weighted merging procedure.

Two different ground precipitation observations, i.e., China Gauge-based Daily Precipitation Analysis (CGDPA; Shen et al., 2016) and the precipitation observations

from more than 30,000 rain gauges, were used here as the benchmark. CGDPA has a 0.25° spatial and daily temporal resolution and its precipitation observations come from ~2400 rain gauges that cover mainland China (Shen et al., 2016). CGDPA is a high-quality ground-based precipitation dataset and has been widely used as the benchmark for the evaluation of various precipitation products (e.g., Ma et al., 2018b; Lyu et a.,



2020; Tang et al., 2020, Chen et al., 2021; Wang et al., 2022; Shaowei et al., 2022).

Thus, CGDPA for the period 2009-2019 (11 years) was used as a reference for

evaluation at a daily scale. Besides, an hourly precipitation dataset from more than

30,000 rain gauges for the period 2014-2018 (5 years) was used as a ground reference

for evaluation at 3 hourly and hourly scales. Those hourly precipitation observations

have been undergone strict quality control (Shen et al., 2014). The spatial distribution

of the rain gauges used in CGDPA and hourly precipitation observations is depicted in

Fig. 2. It is important to note that the evaluation was performed at the grid cells with at

least one rain gauge.

**Insert Fig. 2 about here**

To verify the MPDF algorithm and the performance of the MGP suite products, we

utilized ten error metrics to quantify the performance of the MGP suite products,

including the probability of detection (POD), false alarm ratio (FAR), CC, root mean

squared error (RMSE), normalized mean absolute error (NMAE), normalized RMSE

(NRMSE), total bias and its three independent error components (i.e., hit bias, miss bias,

and false bias).

       Among the ten evaluation metrics, POD and FAR were used to evaluate the

detection capability of precipitation products; CC can accurately describe the

consistency between evaluated precipitation products and gauge observations; RMSE

quantitatively describes the accuracy of evaluated precipitation products; NMAE and

NRMSE were used to evaluate the accuracy of precipitation products at different

rainfall intensity groups; total bias was used to analyze the systematic bias of

precipitation products and reveal the overestimation and underestimation of precipitation amounts for evaluated precipitation products; three independent error components (i.e., hit bias, miss bias, and false bias) were often used to explore the major

error component of the total bias (Tian et al., 2009). The corresponding formulae of the error scores used in this study were provided in Table 1.

**Insert Table 1 about here**

## 3. Results

In this section, the performance of the MPDF algorithm and MGP suite products

was evaluated and verified, including two aspects:

(1) **Global comparison:** a comparison in terms of global patterns of daily mean precipitation between the MGP suite products, MSWEP, IMERG, GSMaP, ERA5, and CPCU was executed to identify the uncertainties of those precipitation products in global land areas.

(2) **Ground verification in mainland China:** we executed the performance evaluation of the MGP suite products at multiple time scales (i.e., daily, 3 hourly, and hourly) to prove the effectiveness of the MPDF algorithm and MGP suite products, using two different ground data as the benchmark. Note that the precipitation accumulation of the daily CGDPA product comes from 8:00 to

8:00, which cannot match with MSWEP (3 hourly temporal resolution). Thus, the MSWEP product was only involved in the comparison at a 3 hourly scale.

### 3.1 Global comparison

Fig. 3 displays global maps of the daily mean precipitation for the seven global precipitation products (i.e., MGP-6P, MGP-3P, MSWEP, IMERG, GSMaP, ERA5, and CPCU). All global products indicate that abundant precipitation appears in the areas closest to the equator and in regions controlled by the monsoon climate, with daily average precipitation amounts greater than 4 mm day$^{-1}$; while the areas with seriously insufficient precipitation include the north and south ends of Africa, Saudi Arabia, central Australia, western China, etc., with daily mean precipitation volumes under 1 mm day$^{-1}$. The results are consistent with Chen et al. (2020b). All global products in terms of daily mean precipitation share high consistency in most global land areas. However, large differences regarding daily mean precipitation between seven global products exist in Australia, southeast China, Europe, near the equator of Africa and South America, and so on. It suggests that those global products have large uncertainties in the above regions, resulting in a large difference in conclusions when using different precipitation products as inputs to investigate local climate change, water cycles, hydrological simulation, and so on (Tang et al., 2020). In general, global pattern of daily mean precipitation for GSMaP is consistent with that for CPCU. A possible reason is that GSMaP was derived from the correction of satellite-only GSMaP-MVK by using CPCU as a reference.

**Insert Fig. 3 about here**



### 3.2  Ground verification in mainland China

### 3.2.1    Overall comparisons of performance

The boxplots of POD and FAR for the six global products (i.e., MGP-6P, MGP-3P,

MSWEP, IMERG, GSMaP, and ERA5) at various time scales (i.e., daily, 3 hourly, and

hourly) are shown in Fig. 4. Obviously, the MGP-3P product in terms of POD has the

best performance at all three time periods. The results are consistent with those of the

summary of POD shown in Table 2, with the highest POD values of 0.85, 0.85, 0.76 for

daily, 3 hourly, and hourly resolutions, respectively. Nevertheless, the worst

performance in terms of FAR falls in the MGP suite products. Similarly, ERA5 has high

POD and FAR values. We speculate that the ERA5 product is the major contributor to

the high POD and FAR values for the MGP suite. This implies that the advantages and

deficiencies of the input data sources will disseminate into merged precipitation

estimates. Accordingly, reducing the false alarms of ERA5 before the weighted merging

procedure is a potentially effective way to remove the false alarms in the MGP suite

products.

**Insert Fig. 4 about here**

**Insert Table 2 about here**

The CC and RMSE metrics can reflect the accuracy of the precipitation products,

Fig. 5 represents the boxplots of CC and RMSE for the six global products (i.e., MGP-

6P, MGP-3P, MSWEP, IMERG, GSMaP, and ERA5) at multiple time scales (i.e., daily,

3 hourly, and hourly). It can see that the MGP-3P product with regard to CC score

performs best at most time scales except for the hourly scale. In fact, the value of CC

for MGP-3P (0.62) at an hourly scale is slightly lower than that for IMERG (0.63) at the same time scale, as shown in Table 2. As for the RMSE metric, the MGP-3P product has the lowest RMSE values at all three time scales, with 5.9 mm on a daily scale, 1.21 mm on a 3-hourly scale, and 0.52 mm on an hourly scale (see Table 2). The results of Table 2 and Fig. 5 show the superiority of MGP-3P in terms of RMSE. Except for the MGP suite products, MSWEP, which is another multi-source fusion product, performs

worse than the MGP suite products and GSMaP (see Table 2 and Fig. 5) at a 3 hourly scale. An explanation might be that the MSWEP algorithm might introduce uncertainties in the weighted merging procedure.

**Insert Fig. 5 about here**

The boxplots of the total bias and its three independent error components (i.e., hit

bias, miss bias, and false bias) are illustrated in Fig. 6. The best product in terms of the total bias varies with the time scale, however, the best one in terms of the total bias is not necessarily the best one in all the three error component cases. The MGP suite products (i.e., MGP-6P and MGP-3P) have the lowest miss biases at each time scale as they have the best detection capability at the corresponding time scale (see Fig. 4). In

practice, the lowest miss bias of the MGP suite products is not conducive to obtaining a lower total bias because a small miss bias will be in an unfavorable position in canceling positive hit bias and false bias (see the relationship between total bias and its three independent error components: $\text{total bias} = \text{hit bias} - \text{miss bias} + \text{false bias}$; Tian et al., 2009). GSMaP has the lowest total bias at both daily and hourly

scales, and MSWEP for a 3 hourly scale. GSMaP comes from the correction of the daily

total precipitation amount of GSMaP-MVK by using CPCU as a reference (Mega et al., 2019), and MSWEP was calculated by temporal downscaling of high-quality ground-based precipitation observations (Beck et al., 2017, 2019). Those might be the reasons for the lowest total bias of GSMaP and MSWEP. In contrast, the MGP suite products

are directly computed by the weighted merging of satellite, ERA5 reanalysis, and ground-based CPCU data, without correcting the total precipitation volumes of satellite and ERA5 reanalysis before the weighted merging procedure. This is a reason that the MGP suite products have larger total biases than GSMaP and MSWEP.

**Insert Fig. 6 about here**

**3.2.2    Spatial comparisons of performance**

Spatial maps of POD for the six global products (i.e., MGP-6P, MGP-3P, MSWEP, IMERG, GSMaP, and ERA5) at three time periods (i.e., daily, 3 hourly, and hourly) are shown in Fig. 7. The MGP suite products perform better than other products at all three time scales over most areas, with higher POD values exceeding 0.7; while among the

fusion products, MGP-3P exhibits the best performance than MGP-6P and MSWEP. It should be noted that the performance of ERA5 in terms of POD at a daily scale is better than that of the MGP suite products at the same time scale in southern areas. Several studies found that satellite precipitation products have a worse detection capability than ERA5 (Jiang et al., 2021; Xu et al., 2022). The deficiencies and limitations of satellite

precipitation retrievals in detection capability were propagated to merged precipitation estimates in the weighted merging procedure, resulting in the performance of the MGP

suite products with regard to POD being worse than ERA5.

**Insert Fig. 7 about here**

Fig. 8 shows the spatial patterns of FAR for the six global products (i.e., MGP-6P,

MGP-3P, MSWEP, IMERG, GSMaP, and ERA5) at three time scales (i.e., daily, 3

hourly, and hourly) over mainland China. One can see that those six global products

show similar spatial patterns in the FAR score at all three time periods, with lower FAR

values in southern areas but higher ones in the remaining areas. The fact is that the

southern areas have a large number of rainfall occurrences and precipitation products

tend to capture a high quantity of rainfall events in such areas. That is a reason why

those six global products have lower FAR values in southern areas (see definition

equation of FAR in Table 1). Overall, IMERG performs best while ERA5 is the worst.

The products with a high POD seem to have a high FAR (see Figs. 7-8), indicating that

the products have a high POD, but that will be at the expense of a high FAR.

**Insert Fig. 8 about here**

The spatial maps of CC for six evaluated global products (i.e., MGP-6P, MGP-3P,

MSWEP, IMERG, GSMaP, and ERA5) at three time scales (i.e., daily, 3 hourly, and

hourly) are shown in Fig. 9. MGP-3P in terms of CC has the best performance in most

areas, with CC values exceeding 0.6. As a multi-source precipitation fusion product,

MSWEP performs worse than other products (except ERA5), suggesting that the

improvement of MSWEP in terms of CC is limited, which might be because of the

algorithm-based errors and uncertainties being introduced in MSWEP estimates; while

ERA5 has the worst performance, with the lowest CC values for all three time periods.



This might be the fact that ERA5 has not been corrected by gauge observations.

**Insert Fig. 9 about here**

Regarding the RMSE score, the differences between six global products occur in southeastern China, as shown in Fig. 10 (at a 3 hourly scale) and supplementary material Figs. S1-S2 (at daily and hourly scales). Meanwhile, six global products have large RMSE in those areas with abundant precipitation, owing to RMSE depending on 515 rainfall amount (Chen et al., 2020b). Overall, the MGP-3P product performs better than the other five products, while the ERA5 product is the worst one at the sub-hourly scale but the daily scale for the IMERG product. This result is consistent with the summary of RMSE for those six global products (see Table 2).

**Insert Fig. 10 about here**

### 3.2.3 Performance comparisons under different rainfall intensity groups

To further diagnose the performance of the MGP suite products and other three global precipitation products (i.e., IMERG, GSMaP, and ERA5), a performance comparison from different rainfall intensity groups was designed here. Hourly precipitation was classified by setting fixed thresholds of 0.2, 0.4, 0.6, 1, 2, and 5 mm 525 as documented in many studies (e.g., Wang et al., 2018; Chen et al., 2020b, 2022b). Note that rainfall intensity categories come from gauge observations, and hence false alarms do not exist.

Fig. 11 depicts the POD, NMAE, and NRMSE of the five global products (i.e., MGP-6P, MGP-3P, IMERG, GSMaP, and ERA5) for the six rainfall intensity groups.



Fig. 11(a) clearly shows that the MGP suite products have higher POD than other

products for all rainfall intensity groups, suggesting that the MPDF algorithm is

effective in improving the detection capability for all rainfall intensity groups. Among

the MGP suite, MGP-3P has a better performance than MGP-6P. In terms of accuracy

metrics (i.e., NMAE and NRMSE), the MGP suite products have lower values of

NMAE and NRMSE than other products at most ranges of rainfall intensities except for

greater than 5 mm hr$^{-1}$. The results suggest that the MPDF algorithm has limitations in

improving the precipitation accuracy at such rainfall intensities, although the detection

capability of the MGP suite products is evidently improved.

**Insert Fig. 11 about here**

The histograms of the total bias and its two independent error components of the

five global products (i.e., MGP-6P, MGP-3P, IMERG, GSMaP, and ERA5) for the six

rainfall intensity groups are presented in Fig. 12. The best product for different rainfall

intensity groups in terms of the total bias is different. The MGP suite products have a

lower hit bias than other products at rainfall intensities less than 1 mm hr$^{-1}$, while a

larger hit bias at rainfall intensities greater than 1 mm hr$^{-1}$. It is worth noting that the

MGP suite products exhibit lower miss biases than other global products at all rainfall

intensity groups. This is attributed to the fact that the MGP suite products are good at

detecting surface rainfall occurrences at all rainfall intensity groups (see Fig. 11a).

**Insert Fig. 12 about here**



## 4. Discussions

### 4.1 Impact of the number of input sources on the quality of the merged estimates

There is an interesting question whether the more precipitation data are merged, the better the merged precipitation estimates will be. The solution to this question is conducive to providing constructive suggestions for the design of the multi-source precipitation data fusion algorithms. Given the satellite-only IMERG-Late product exhibits better performance than other satellite-only precipitation products (Tang et al., 2020; Chen et al., 2020b, 2021), and ERA5 has acceptable detection capability and performs better in the winter season and high-latitude areas (Beck et al., 2017, 2019; Tang et al., 2020; Xu et al., 2022). Spatial maps of the weights designed to satellite, reanalysis, and CPCU also confirm satisfactory performance of ERA5 in the winter season and high-latitude areas, as shown in Fig. 13 and supplementary material Figs. S3-S6. Consequently, we only selected IMERG-Late, ERA5, and CPCU as input data sources in scheme 2 to produce MGP-3P. A performance comparison between MGP-3P and MGP-6P was designed to answer the above question.

Table 3 offers a difference percentage in terms of POD, FAR, CC, and RMSE between MGP-3P and MGP-6P. The results clearly show that MGP-3P has a minor improvement for MGP-6P in all four metrics (i.e., POD, FAR, CC, and RMSE). It means that the number of input data sources is not directly proportional to the quality of the merged precipitation estimates. The quality of the selected input data sources might be the most crucial factor, not the more input data sources the better. The results

from the histogram of the performance differences between MGP-3P and MGP-6P at different rainfall intensity groups also support the above conclusions, shown in Fig. 14.

**Insert Fig. 13 about here**

**Insert Table 3 about here**

**Insert Fig. 14 about here**

## 4.2 Impact of the temporal downscaling on the performance

To generate a global gridded precipitation product with a high temporal resolution, temporal downscaling is an indispensable step in the data merging procedure. Tan et al. (2017) found that the quality of satellite precipitation products decreases when scaled 580 down to shorter time periods. As one of the modules for the MPDF algorithm, the effectiveness of a temporal downscaling technique used plays a qualitative role in the quality of merged precipitation estimates to some extent.

The decay rates of the five global products (i.e., MGP-6P, MGP-3P, IMERG, GSMaP, and ERA5) in terms of POD, FAR, and CC from a 3 hourly resolution scaling 585 down to an hourly resolution are given in Table 4. RMSE is not used here for analysis in that it is affected by rainfall accumulation amount (Chen et al., 2020b). The effect of temporal downscaling on the performance of each global precipitation product for different scores is different. The decay rates of POD and CC exceed 10% in most cases when a 3 hourly resolution is scaled down to an hourly resolution, while ones of FAR 590 are between 0% and 6%. This result indicates that there is a performance degradation for the five research-quality global products to some extent when a 3 hourly resolution

is scaled down to an hourly resolution. Improving the temporal downscaling module used in the MPDF algorithm is one of the potential directions for advancing the quality of MGP suite products.

**Insert Table 4 about here**

### 4.3 Future improvements of the MGP-3P product

Overall, MGP-3P performs better than the other six high-quality global precipitation products (i.e., MGP-6P, MSWEP, IMERG, IMERG, GSMaP, and ERA5), which has been proven by ground verification in mainland China. Nevertheless, there are several

limitations that could be improved for MGP-3P in the future.

**Spatiotemporal resolutions and temporal coverage:** a reliable global precipitation product with finer spatiotemporal resolutions and long-term temporal coverage is critical for a variety of science communities (e.g., hydrology, meteorology, climatology, ecology, and agriculture; Hou et al., 2014; Maggioni et al., 2016; Ma et al.,

2020). To the best of our knowledge, among the (quasi-)global precipitation products, the gauge-based WorldClim product (Fick and Hijmans, 2017) has the finest spatial resolution of ~ 1 km, while the satellite-based PERSIANN-CCS product (Hong et al., 2004) has a fine spatial resolution of ~ 0.04°; for temporal resolution, the satellite-based IMERG family products (Huffman et al., 2019) have the finest temporal resolution of

30 min. However, MGP-3P is generated at a 0.25° spatial and hourly temporal resolution, by using the MPDF algorithm to merge satellite-based IMERG-Late, ERA5 reanalysis, and ground-based CPCU data. In addition, the MGP-3P product has only

been available for the period from 2000 to 2020. Thus, the spatiotemporal resolutions of MGP-3P should be further finer, and its temporal coverage should be extensive for a

longer period, in the next version.

**Reducing the total bias and false alarms:** the evaluation results from mainland China indicated that MGP-3P has an unsatisfactory performance in total bias and false alarms. We speculate that the large false bias of MGP-3P is mainly from ERA5 because ERA5 has higher false alarms, and these false alarms were propagated to merged

precipitation estimates in the weighted merging procedure. A way using satellite soil moisture or integrating cloud properties as auxiliary information might be effective in reducing false alarms in ERA5 as this method has been proven to effectively remove false alarms in satellite precipitation products (e.g., Crow et al., 2011; Wanders et al., 2015; Massari et al., 2019; Zhang et al., 2021), so as to remove the false alarms of

MGP-3P to some extent. Then, using ground precipitation observations to correct the rainfall amount of the satellite-derived IMERG-Late and false-corrected ERA5 reanalysis before the weighted merging procedure might be effective in reducing the total bias of MGP-3P.

**Extension of area coverage:** the current version of MGP-3P is a global land

precipitation product without precipitation information on the ocean as the weighting method used in this study depends on the ground-based CPCU data as a reference. Precipitation information on the ocean is critical for various science communities. An effective weighting method without using ground-based precipitation observations as a reference will be developed to obtain the weights of IMERG-Late and ERA5 in

ungauged areas especially for the ocean areas, for producing a new version of the MGP-

3P product with covering precipitation information on the ocean.

**Limitations in performance evaluation:** we executed a global land comparison in

daily mean precipitation between seven high-quality global products, including MGP-

6P, MGP-3P, MSWEP, IMERG, GSMaP, ERA5, and CPCU, to reveal the uncertainties

of precipitation estimates for those global precipitation products. A large uncertainty of

the seven global precipitation products occurs in Australia, southeast China, Europe,

near the equator of Africa and South America, and so on. This finding highlights that a

detailed evaluation of the seven global precipitation products in above areas should be

executed as soon as possible. In addition, a quantitative performance evaluation was

executed at three time periods (i.e., daily, 3 hourly, and hourly) over mainland China by

using two different high-quality ground precipitation observations as the benchmark.

The temporal coverage of the evaluation for mainland China is from 2009 to 2019 on a

daily scale and from 2014 to 2019 on a 3 hourly scale and an hourly scale. Although

preliminary evaluation results have verified the performance of the MGP-3P product to

some extent, more evaluation efforts on a longer temporal coverage and global scale

are necessary to reveal the advantages and limitations of the MGP-3P product in detail.

Because of the limited high-quality independent ground precipitation observations

available to us, we hope researchers around the world can implement more detailed

evaluations and applications for the MGP-3P product, providing more detailed error

features for us to update the MPDF algorithm to improve the quality of MGP-3P.



## 5. Data availability

The MGP-3P (also known as MGP) Version 1.1 is available at the following link: https://www.zenodo.org/record/7386441#.Y8zr4clBxD9 (Chen et al., 2022a).

## 6. Conclusions

Precipitation is one of the main components of the global water cycle and global energy cycle, as well as the main input for various hydrological simulations. Accurate precipitation estimates are essential for the above studies and operational applications. To obtain reliable precipitation estimates, we proposed a novel multi-source precipitation data fusion algorithm with considering the impact of seasonality on
precipitation errors to take advantage of the complementary strengths of different precipitation data sources (i.e., satellite, reanalysis, and ground-based precipitation observations) to generate a higher-quality global precipitation product. Two different merging schemes were designed to investigate the effect of the number of input data sources on the quality of merged precipitation estimates. The main conclusions and
findings are summarized as follows:

1. Seven global precipitation products (i.e., MGP-6P, MGP-3P, MSWEP, IMERG-Final, GSMaP-Gauge, ERA5, and CPCU) share a similar spatial pattern in the daily mean precipitation in most land areas of the world. However, their differences occur in Australia, southeast China, Europe, near the equator of
Africa and South America, and so on, indicating that the precipitation estimates from those global precipitation products in such areas have large uncertainties.

It implies that different conclusions might exist when using different precipitation products as input to explore local climate change analysis, hydrological simulation, water cycle, and so on. The finding highlights that the

performance evaluation of global precipitation products in such areas should be valued.

2. Six global precipitation products (i.e., MGP-6P, MGP-3P, MSWEP, IMERG-Final, GSMaP-Gauge, and ERA5) were evaluated at three time periods (i.e., daily, 3 hourly, and hourly) in mainland China using two different ground-based

precipitation datasets as the benchmark. The MGP-3P product in terms of most metrics performs better than the other five global precipitation products, with 0.85, 0.85, and 0.76 for POD, 0.70, 0.71, and 0.62 for CC, and 5.90 mm, 1.21 mm, and 0.52 mm for RMSE, for three time periods, respectively. However, MGP-3P does not perform best in FAR and total bias, because of the propagation

of false alarms for ERA5 to merged precipitation estimates and without correcting the total precipitation amounts of satellite- and reanalysis-based products before the weighted merging procedure.

3. MGP-3P has a higher (lower) POD (NMAE) than other evaluated global products at all rainfall intensity groups; while the NRMSE of MGP-3P is

slightly higher than that of gauge-adjusted IMERG-Final at the ranges of rainfall intensities exceeding 5 mm hr$^{-1}$. In terms of error components, MGP-3P has a lower miss bias in all rainfall intensity groups as it has the best detection capability. Nevertheless, a low miss bias is limited in canceling positive hit bias,

which leads to a large total bias of MGP-3P at most rainfall intensity groups.

4. We found that the quality of input data sources is critical for that of merged

precipitation estimates, not the more input data sources the better. Additionally,

the impact of temporal downscaling on different metrics is different. Overall,

the performance of global precipitation products decreases when scaled down

to a shorter time period. In particular, the performance reduction of all evaluated

global precipitation products in terms of POD and CC exceeds 7% when a 3

hourly scale is scaled down to an hourly scale, indicating that the temporal

downscaling modules used in those global products need to be further improved.

Our initial evaluation results indicated that the MPDF algorithm is effective in

considering the advantages from different precipitation data sources; overall, the MGP-

3P product performs better than the other five products (i.e., MGP-6P, MSWEP,

IMERG-Final, GSMaP-Gauge, and ERA5). Yet, the current version of the MGP-3P

product still needs further improvements in multiple aspects, such as reducing the total

bias and false alarms, advancing the spatiotemporal resolutions, and extending the

temporal coverage and area coverage. Finally, we hope MGP-3P will play an important

role in scientific research and various operational applications.

## Author contributions

HC was responsible for the conceptualization, methodology, software, formal

analysis, writing, and funding acquisition. DW was responsible for the

conceptualization, writing-review and editing, project administration, and funding

acquisition. BY was responsible for the conceptualization, writing-review and editing, project administration, and funding acquisition. JJG was responsible for writing-review and editing and methodology. LW was responsible for writing-review and editing. YH was responsible for the methodology and conceptualization.

## Competing Interests

The authors declare that they have no conflict of interest.

## Acknowledgments

We are very grateful to the developers of global precipitation products and ground observations used in this work.

## Financial support

This work was mainly sponsored by the National Natural Science Foundation of China (grant nos. 42201029, 42074030), and was supported in part by the National Key Research and Development Program of China (grant no. 2018YFA0605402) and in part by the China Postdoctoral Science Foundation (grant no. 2021M700923).

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


**Table and Figure captions**

**Table 1** Ten error metrics used in this study.

**Table 2** Summary of the four evaluation scores (i.e., POD, FAR, CC, and RMSE) for the six global precipitation products (i.e., MGP-6P, MGP-3P, MSWEP, IMERG, GSMaP, and ERA5) at three time periods (i.e., daily, 3 hourly, and hourly) over mainland China. Note that the product corresponding to the shadow is the best one for each evaluation metrics.

**Table 3** The difference percentages in terms of POD, FAR, CC, and RMSE between MGP-3P and MGP-6P.

**Table 4** Decay rates of the five products (i.e., MGP-6P, MGP-3P, IMERG, GSMaP, and ERA5) in terms of POD, FAR, and CC when from a 3 hourly resolution scaling down to an hourly resolution.

**Fig. 1** Flowchart of the multi-source precipitation data fusion (MPDF) algorithm proposed in this study.

**Fig. 2** Spatial maps of the rain gauges used in this study: (a) CPCU; (b) CGDPA; (c) hourly precipitation observations.

**Fig. 3** Global land maps of the daily average precipitation for the seven global 1115 precipitation products (i.e., MGP-6P, MGP-3P, MSWEP, IMERG, GSMaP, ERA5, and CPCU).

**Fig. 4** Boxplots of POD and FAR for the six global precipitation products (i.e., MGP-6P, MGP-3P, MSWEP, IMERG, GSMaP, and ERA5) at three time periods (i.e., daily, 3 hourly, and hourly) for the whole Chinese mainland.



**Fig. 5** Boxplots of CC and RMSE for the six global precipitation products (i.e., MGP-6P, MGP-3P, MSWEP, IMERG, GSMaP, and ERA5) at three time periods (i.e., daily, 3 hourly, and hourly) for the whole mainland China.

**Fig. 6** Boxplots of the total bias and its three independent error components (i.e., hit bias, miss bias, and false bias) for the six global precipitation products (i.e., MGP-6P, 1125 MGP-3P, MSWEP, IMERG, GSMaP, and ERA5) at three time periods (i.e., daily, 3 hourly, and hourly) for the whole mainland China.

**Fig. 7** Spatial maps of POD for the six global precipitation products (i.e., MGP-6P, MGP-3P, MSWEP, IMERG, GSMaP, and ERA5) at three time periods (i.e., daily, 3 hourly, and hourly) over Chinese mainland.

**Fig. 8** Spatial maps of FAR for the six global precipitation products (i.e., MGP-6P, MGP-3P, MSWEP, IMERG, GSMaP, and ERA5) at three time periods (i.e., daily, 3 hourly, and hourly) over mainland China.

**Fig. 9** Spatial maps of CC for the six global precipitation products (i.e., MGP-6P, MGP-3P, MSWEP, IMERG, GSMaP, and ERA5) at three time periods (i.e., daily, 3 hourly, 1135 and hourly) over mainland China.

**Fig. 10** Spatial maps of RMSE for the six global precipitation products (i.e., MGP-6P, MGP-3P, MSWEP, IMERG, GSMaP, and ERA5) at a 3 hourly scale over mainland China.

**Fig. 11** Histograms of POD, NMAE, and NRMSE of the five global precipitation 1140 products (i.e., MGP-6P, MGP-3P, IMERG, GSMaP, and ERA5) at the six hourly rainfall intensity groups for the whole mainland China.



**Fig. 12** Histograms of the total bias and its two error components (i.e., hit bias, and false bias) of the five global precipitation products (i.e., MGP-6P, MGP-3P, IMERG, GSMaP, and ERA5) at the six hourly rainfall intensity groups for the whole mainland China. Note that there is no false bias because rainfall intensity groups come from the benchmark.

**Fig. 13** Global land maps of the weights designed to satellite, reanalysis, and CPCU in four seasons (i.e., MAM, JJA, SON, and DJF) for scheme 2.

**Fig. 14** Histograms of the performance differences in terms of POD, NMAE, and NRMSE between MGP-3P and MGP-6P at the six rainfall intensity groups for the whole mainland China.



**Table 1** Ten error metrics used in this study.

| Error metrics | Equations | Perfect value |
|---|---|---|
| Probability Of Detection (POD) | $\text{POD} = \dfrac{H}{H + M}$ | 1 |
| False Alarm Ratio (FAR) | $\text{FAR} = \dfrac{F}{H + F}$ | 0 |
| Correlation Coefficient (CC) | $\text{CC} = \dfrac{\sum_{i=1}^{n}(G_i - \bar{G})(S_i - \bar{S})}{\sqrt{\sum_{i=1}^{n}(G_i - \bar{G})^2} \times \sqrt{\sum_{i=1}^{n}(S_i - \bar{S})^2}}$ | 1 |
| Root Mean Squared Error (RMSE) | $\text{RMSE} = \sqrt{\dfrac{1}{n}\sum_{i=1}^{n}(S_i - G_i)^2}$ | 0 |
| Normalized Mean Absolute Error (NMAE) | $\text{NMAE} = \dfrac{\sum_{i=1}^{n}|S_i - G_i|}{\sum_{i=1}^{n} G_i}$ | 0 |
| Normalized RMSE (NRMSE) | $\text{NRMSE} = \dfrac{\sqrt{\dfrac{1}{n}\sum_{i=1}^{n}(S_i - G_i)^2}}{\bar{G}}$ | 0 |
| Total bias | $\text{Total bias} = \dfrac{\sum_{i=1}^{n}(S_i - G_i)}{\sum_{i=1}^{n} G_i} \times 100\%$ | 0 |
| Hit bias | $\text{Hit bias} = \dfrac{\sum_{i=1}^{n}(S_{H_i} - G_{H_i})}{\sum_{i=1}^{n} G} \times 100\%$ | 0 |
| Miss bias | $\text{Miss bias} = \dfrac{\sum_{i=1}^{n}(-G_{M_i})}{\sum_{i=1}^{n} G} \times 100\%$ | 0 |
| False bias | $\text{False bias} = \dfrac{\sum_{i=1}^{n}(S_{F_i})}{\sum_{i=1}^{n} G} \times 100\%$ | 0 |




**Table 2** Summary of the four evaluation scores (i.e., POD, FAR, CC, and RMSE) for the six global precipitation products (i.e., MGP-6P, MGP-3P, MSWEP, IMERG, GSMaP, and ERA5) at three time periods (i.e., daily, 3 hourly, and hourly) over mainland China. Note that the product corresponding to the shadow is the best one for each evaluation metrics.

| Products | Time scales | POD | FAR | CC | RMSE (mm) |
|---|---|---|---|---|---|
| MGP-6P | | 0.83 | 0.43 | 0.67 | 6.17 |
| MGP-3P | | 0.85 | 0.39 | 0.70 | 5.90 |
| IMERG | Daily | 0.67 | 0.39 | 0.63 | 7.05 |
| GSMaP | | 0.83 | 0.34 | 0.69 | 6.00 |
| ERA5 | | 0.85 | 0.41 | 0.66 | 6.52 |
| MGP-6P | | 0.83 | 0.55 | 0.69 | 1.25 |
| MGP-3P | | 0.85 | 0.55 | 0.71 | 1.21 |
| MSWEP | | 0.83 | 0.55 | 0.65 | 1.29 |
| IMERG | 3 hourly | 0.66 | 0.45 | 0.70 | 1.35 |
| GSMaP | | 0.77 | 0.52 | 0.69 | 1.26 |
| ERA5 | | 0.80 | 0.60 | 0.45 | 1.60 |
| MGP-6P | | 0.72 | 0.55 | 0.60 | 0.55 |
| MGP-3P | | 0.76 | 0.56 | 0.62 | 0.52 |
| IMERG | Hourly | 0.61 | 0.47 | 0.63 | 0.58 |
| GSMaP | | 0.69 | 0.55 | 0.60 | 0.55 |
| ERA5 | | 0.68 | 0.63 | 0.36 | 0.67 |



**Table 3** The difference percentages in terms of POD, FAR, CC, and RMSE between

MGP-3P and MGP-6P.

| Time scales | POD (%) | FAR (%) | CC (%) | RMSE (%) |
|:-----------:|:-------:|:-------:|:------:|:--------:|
| Daily | 2.41 | 9.30 | 4.48 | 4.38 |
| 3 hourly | 2.41 | 0.00 | 2.90 | 3.20 |
| Hourly | 5.56 | 1.81 | 3.33 | 5.45 |




**Table 4** Decay rates of the five products (i.e., MGP-6P, MGP-3P, IMERG, GSMaP, and ERA5) in terms of POD, FAR, and CC when from a 3 hourly resolution scaling down to an hourly resolution.

| Products | POD (%) | FAR (%) | CC (%) |
|----------|---------|---------|--------|
| MGP-6P | 13.25 | 0.00 | 13.04 |
| MGP-3P | 10.59 | 1.81 | 12.68 |
| IMERG | 7.58 | 4.44 | 10.00 |
| GSMaP | 10.39 | 5.77 | 13.04 |
| ERA5 | 15.00 | 5.00 | 20.00 |



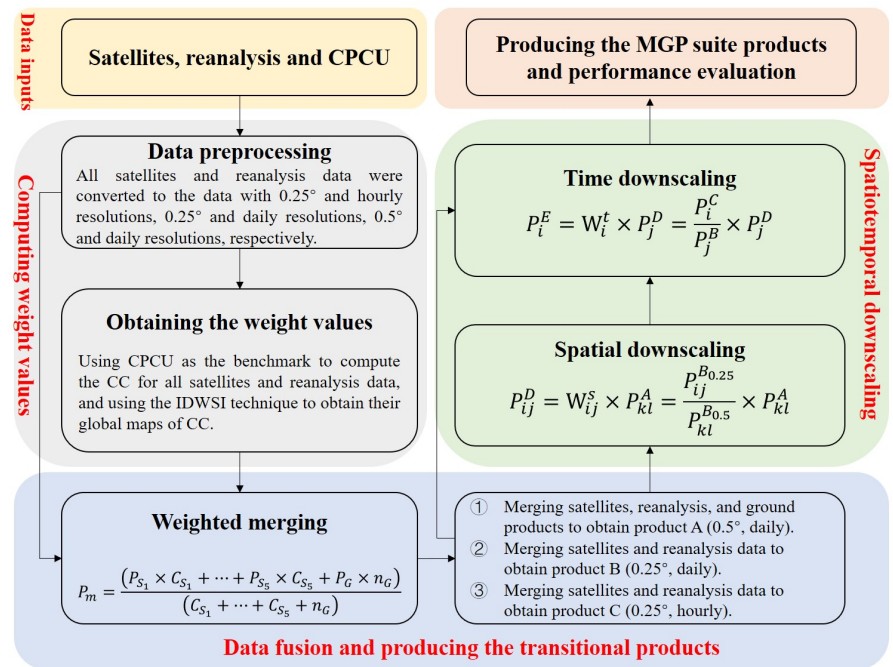


**Fig. 1** Flowchart of the multi-source precipitation data fusion (MPDF) algorithm

proposed in this study.

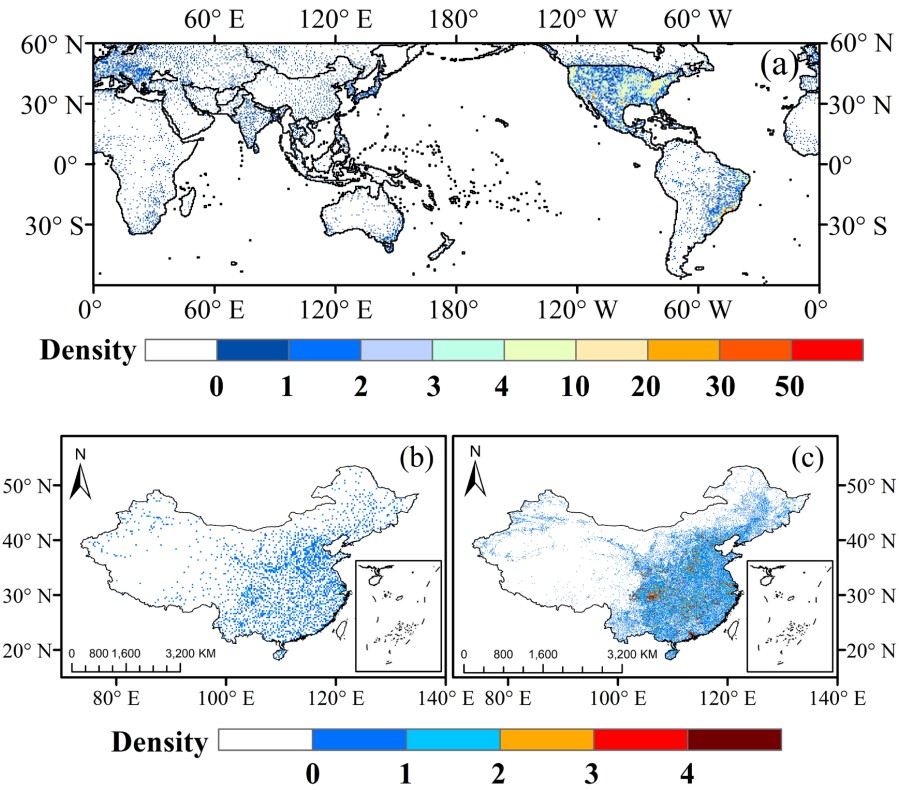

**Fig. 2** Spatial maps of the rain gauges used in this study: (a) CPCU; (b) CGDPA; (c)

hourly precipitation observations.

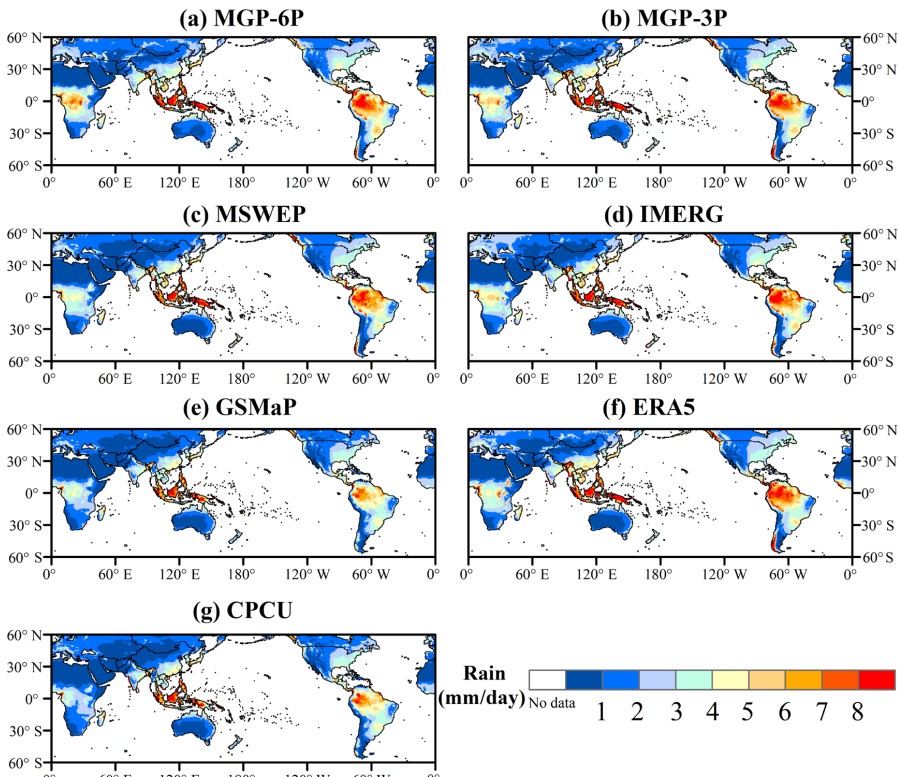

**Fig. 3** Global land maps of the daily average precipitation for the seven global
precipitation products (i.e., MGP-6P, MGP-3P, MSWEP, IMERG, GSMaP, ERA5, and
CPCU).

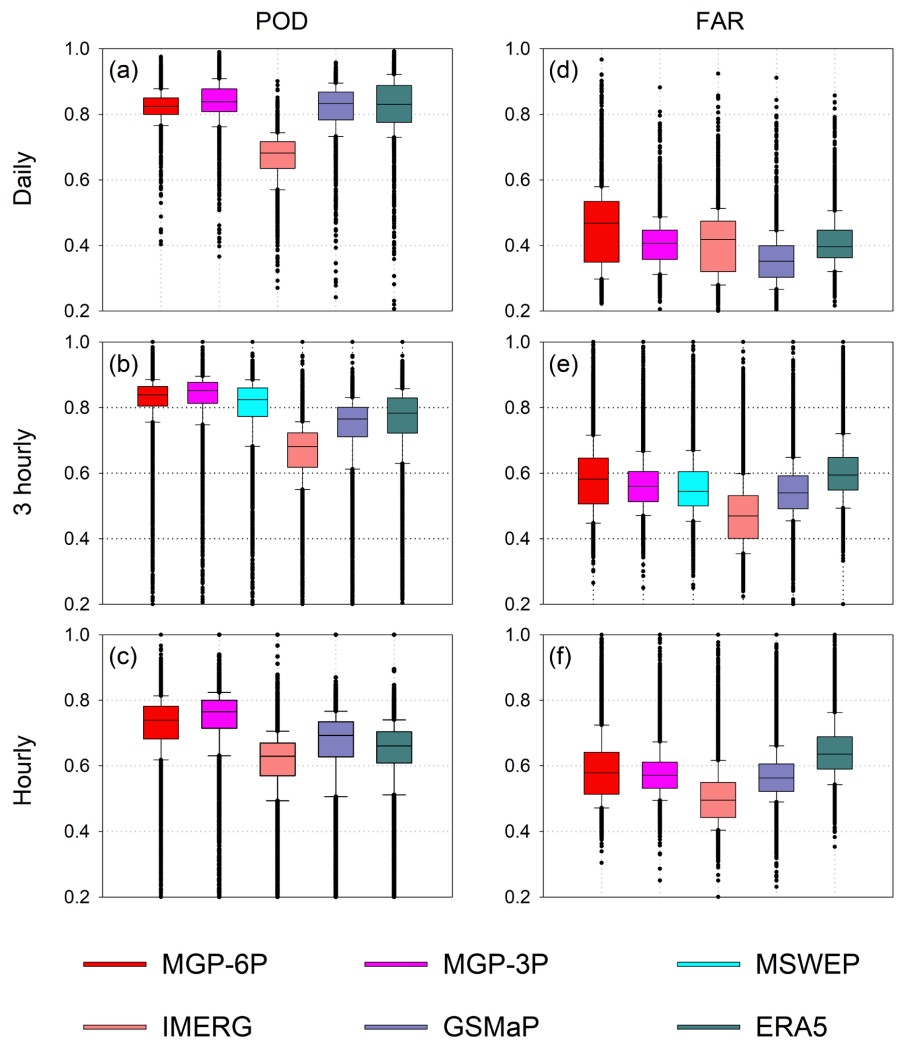

**Fig. 4** Boxplots of POD and FAR for the six global precipitation products (i.e., MGP-

6P, MGP-3P, MSWEP, IMERG, GSMaP, and ERA5) at three time periods (i.e., daily, 3

hourly, and hourly) for the whole Chinese mainland.



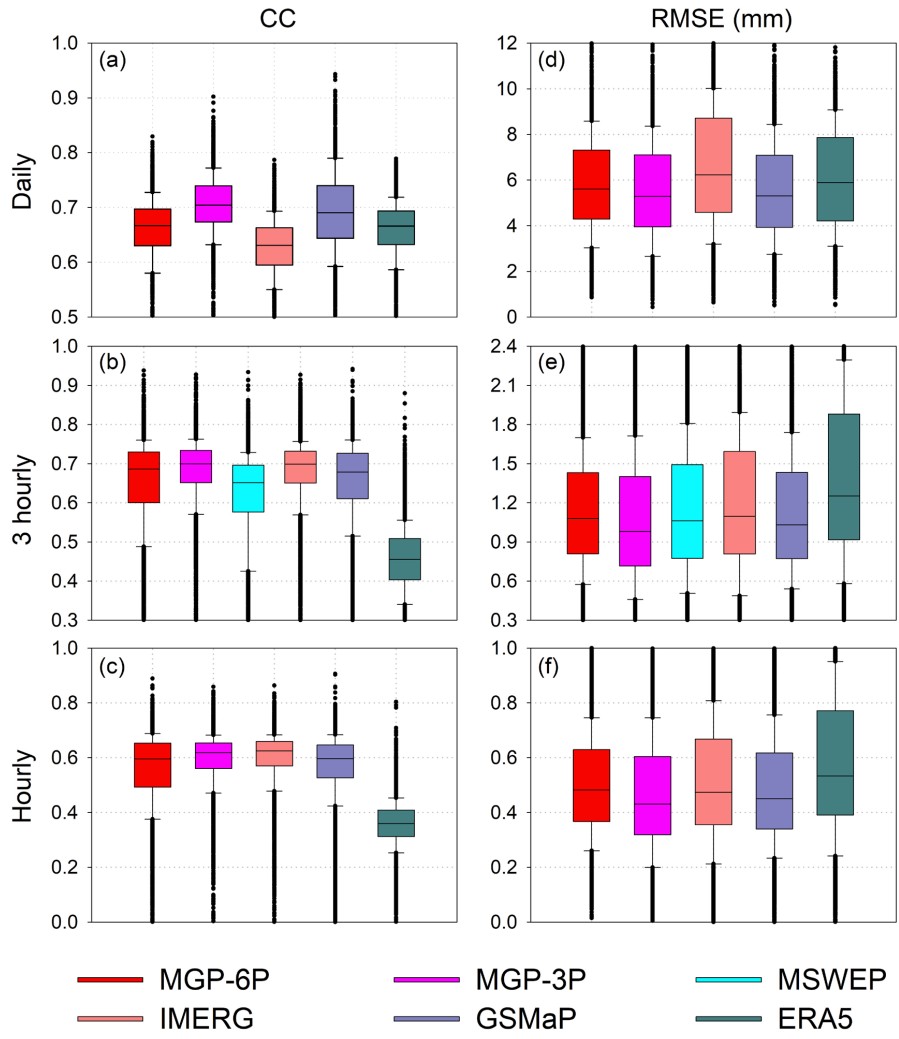

**Fig. 5** Boxplots of CC and RMSE for the six global precipitation products (i.e., MGP-
6P, MGP-3P, MSWEP, IMERG, GSMaP, and ERA5) at three time periods (i.e., daily, 3
hourly, and hourly) for the whole mainland China.

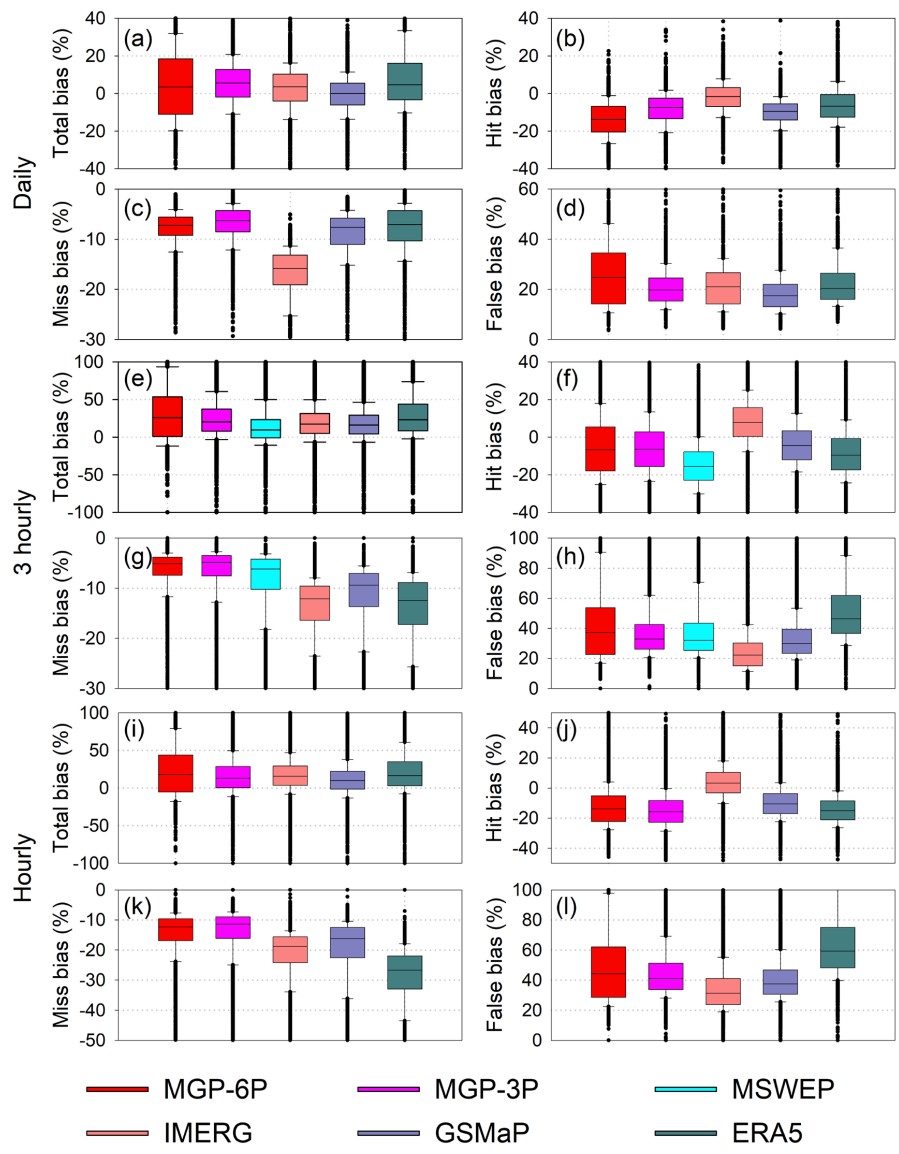

**Fig. 6** Boxplots of the total bias and its three independent error components (i.e., hit

bias, miss bias, and false bias) for the six global precipitation products (i.e., MGP-6P,

MGP-3P, MSWEP, IMERG, GSMaP, and ERA5) at three time periods (i.e., daily, 3

hourly, and hourly) for the whole mainland China.



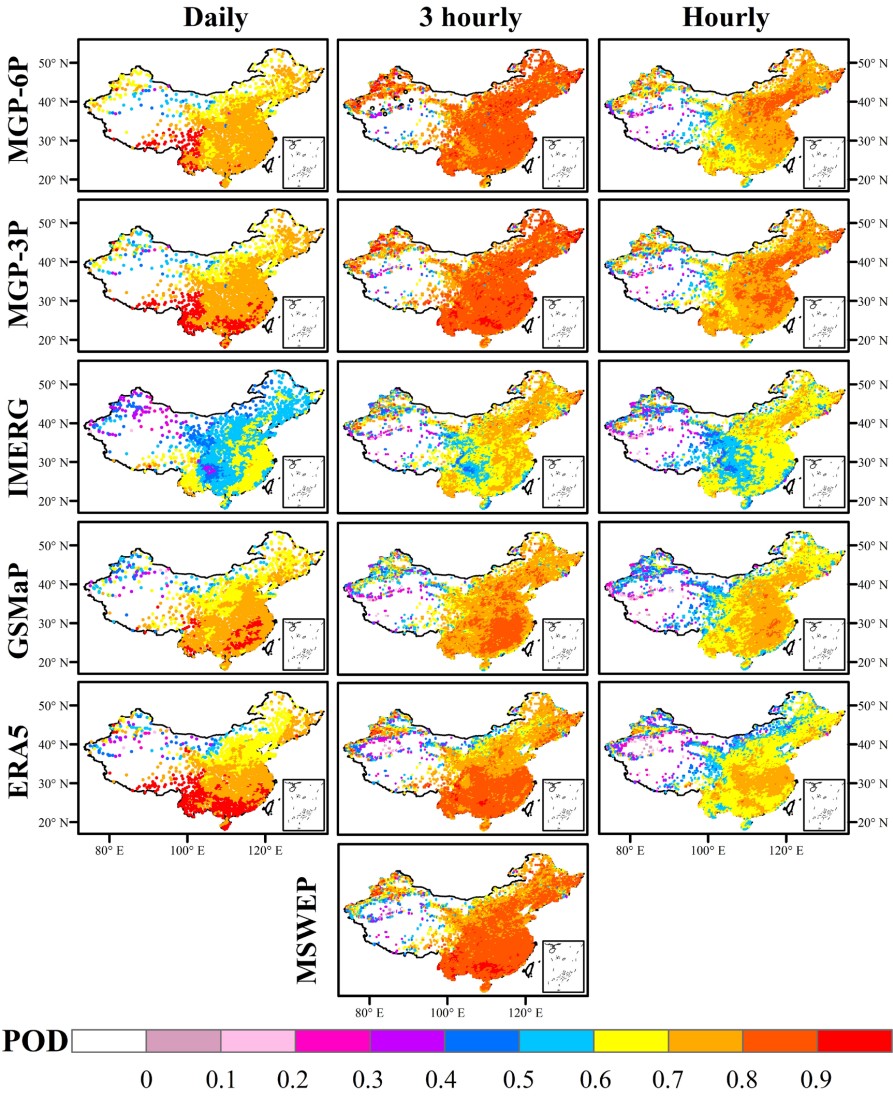

**Fig. 7** Spatial maps of POD for the six global precipitation products (i.e., MGP-6P,

MGP-3P, MSWEP, IMERG, GSMaP, and ERA5) at three time periods (i.e., daily, 3

hourly, and hourly) over Chinese mainland.

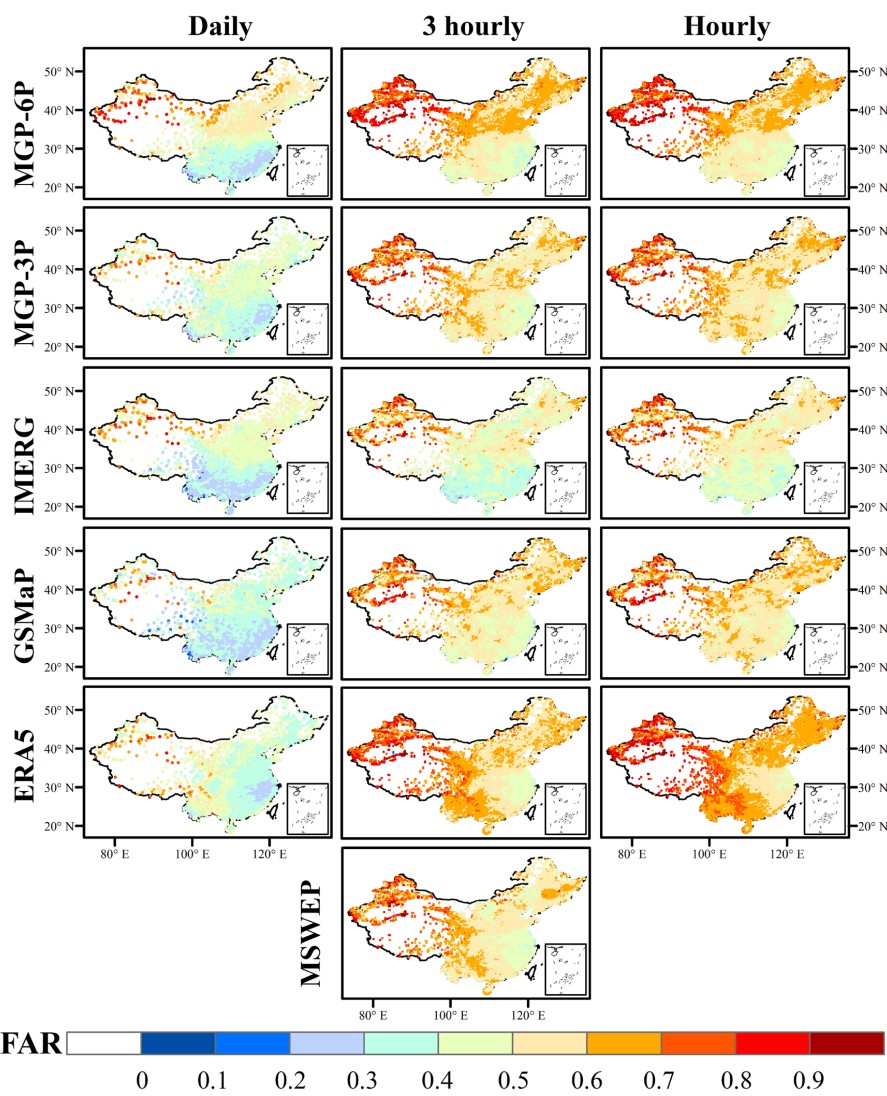

**Fig. 8** Spatial maps of FAR for the six global precipitation products (i.e., MGP-6P, MGP-3P, MSWEP, IMERG, GSMaP, and ERA5) at three time periods (i.e., daily, 3 hourly, and hourly) over mainland China.

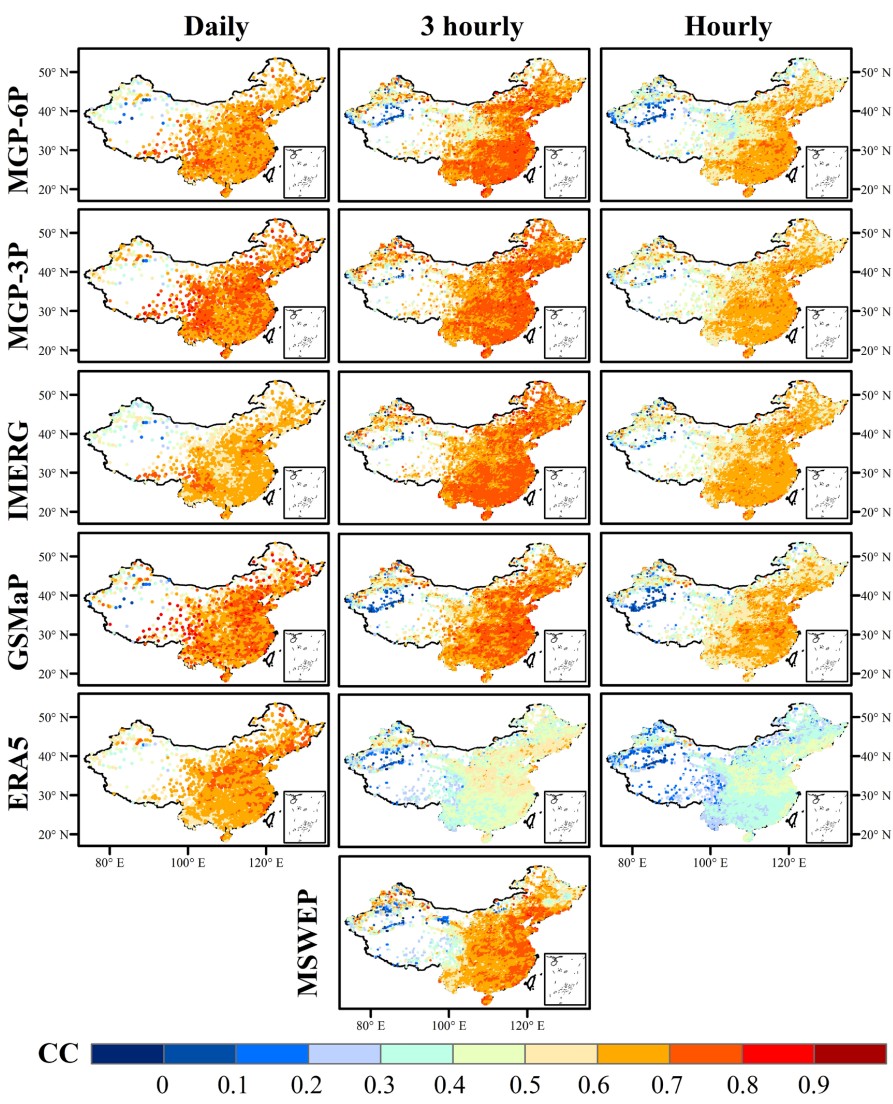

**Fig. 9** Spatial maps of CC for the six global precipitation products (i.e., MGP-6P, MGP-3P, MSWEP, IMERG, GSMaP, and ERA5) at three time periods (i.e., daily, 3 hourly, and hourly) over mainland China.

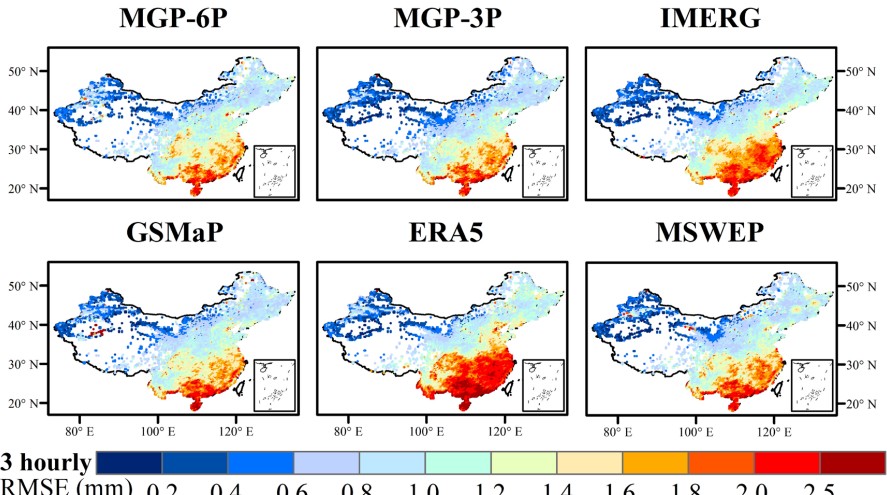

**Fig. 10** Spatial maps of RMSE for the six global precipitation products (i.e., MGP-6P, MGP-3P, MSWEP, IMERG, GSMaP, and ERA5) at a 3 hourly scale over mainland China.

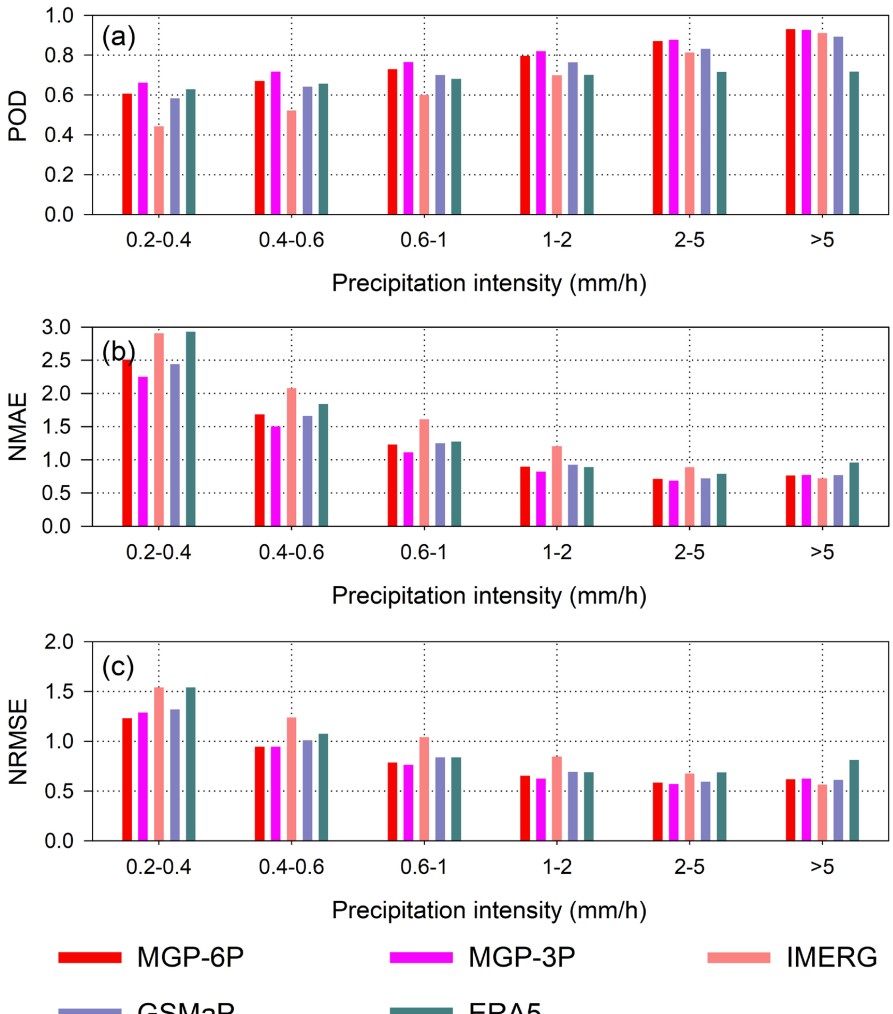

**Fig. 11** Histograms of POD, NMAE, and NRMSE of the five global precipitation

products (i.e., MGP-6P, MGP-3P, IMERG, GSMaP, and ERA5) at the six hourly rainfall

intensity groups for the whole mainland China.



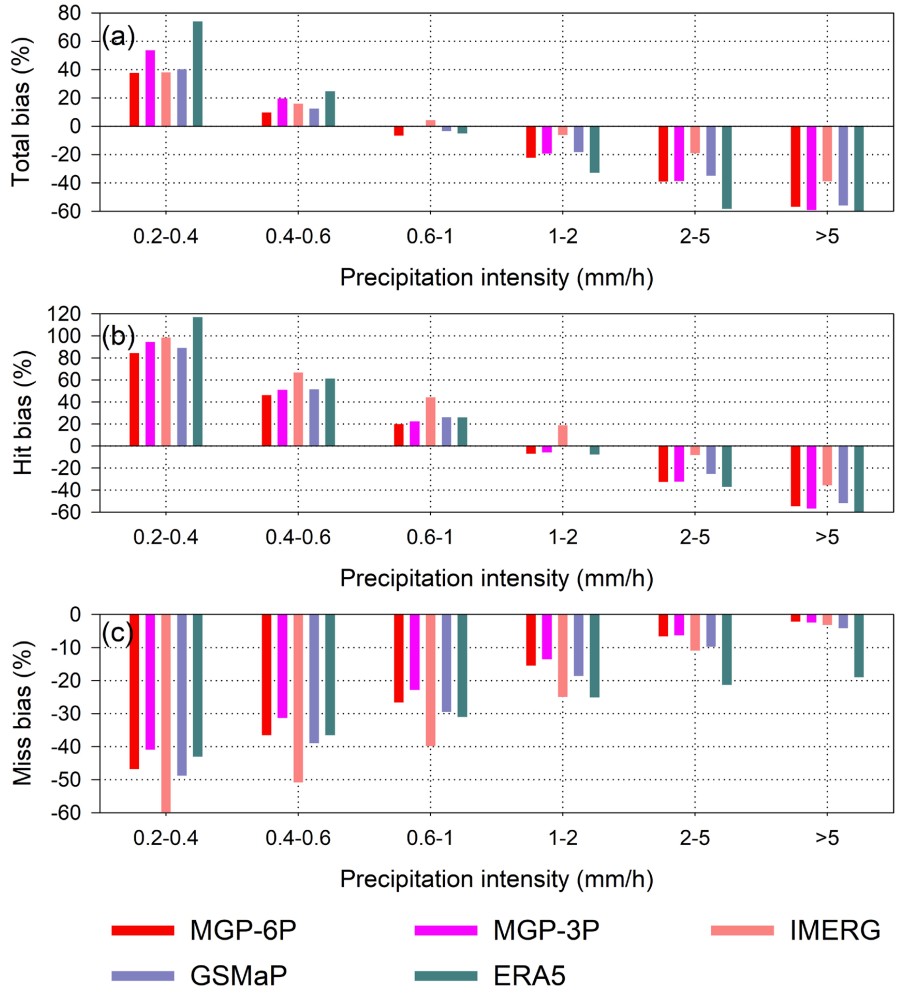

**Fig. 12** Histograms of the total bias and its two error components (i.e., hit bias, and false bias) of the five global precipitation products (i.e., MGP-6P, MGP-3P, IMERG, GSMaP, and ERA5) at the six hourly rainfall intensity groups for the whole mainland China. Note that there is no false bias because rainfall intensity groups come from the benchmark.





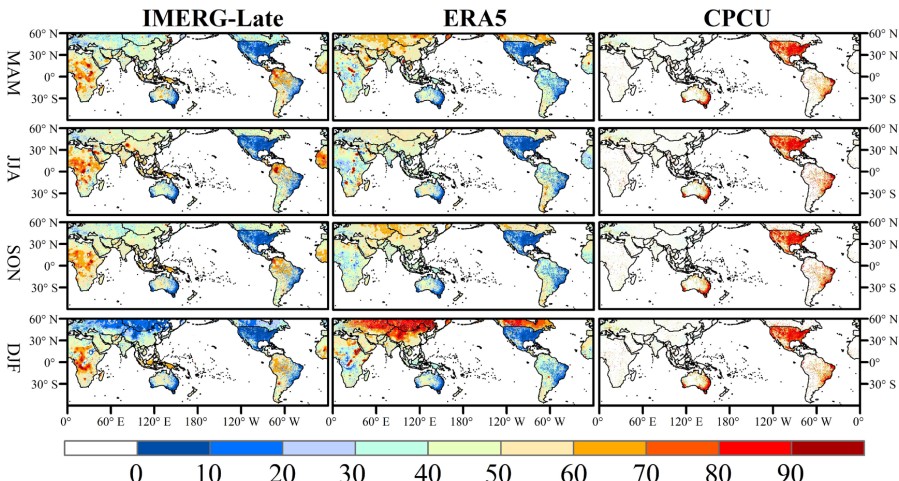

**Fig. 13** Global land maps of the weights designed to satellite, reanalysis, and CPCU in

four seasons (i.e., MAM, JJA, SON, and DJF) for scheme 2.

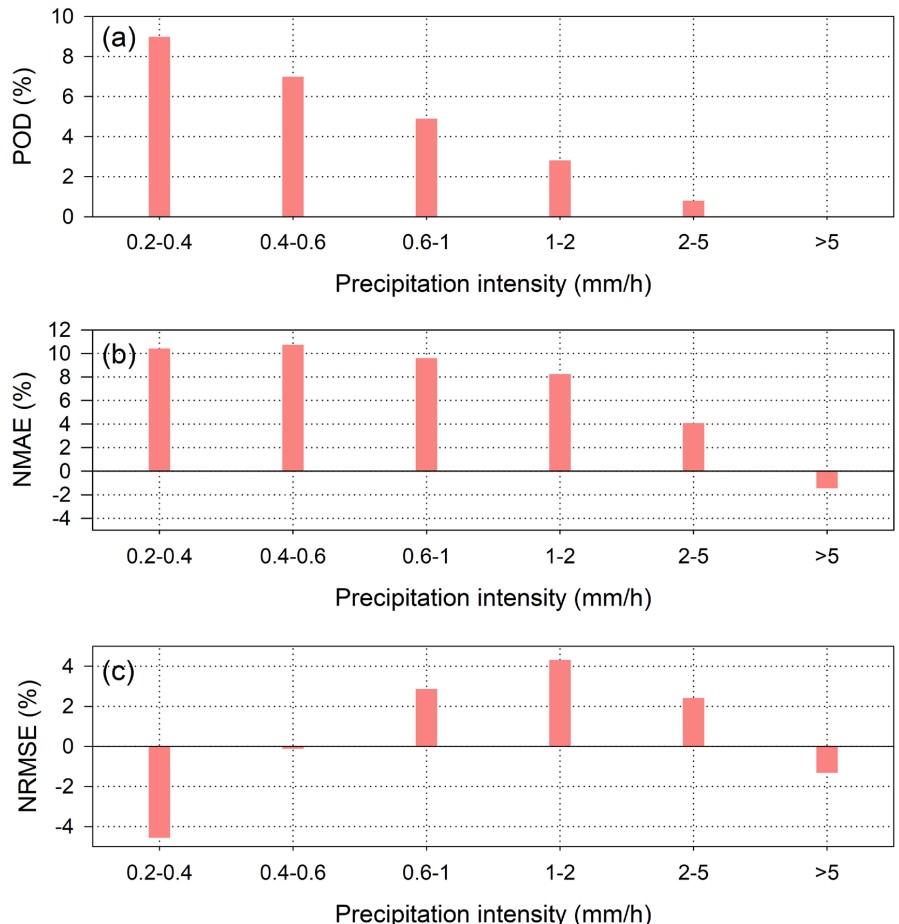


**Fig. 14** Histograms of the performance differences in terms of POD, NMAE, and

NRMSE between MGP-3P and MGP-6P at the six rainfall intensity groups for the

whole mainland China.