# Peer review of "MGP: a new 1-hourly 0.25° global precipitation product (2000-2020) based on multi-source precipitation data fusion Hanqing Chena, Debao Wena\*, Bin Yongb\*, Jonathan J. Gourleyc, Leyang Wangd, Yang Honge"

_Earth System Science Data, 2023_

## Author Comment (AC1)

**Response for Reviewer #1**

High-quality global precipitation product with finer spatiotemporal resolutions and long-term temporal coverage is really critical for a variety of science communities. However, after carefully reading this manuscript, there are various aspects confusing me a lot. Most critically, the writing and organization are really too weak to understand the key ideas of this study, as well as lacking scientific innovative contributions for the community, which seems to be just mixing several global precipitation datasets without any clear new thoughts. Some more serious scientific issues could be seen as follows. Considering the high standards of the big journal, ESSD, I think this study have great limitations and too far distances from the standards.

Response: We are very grateful to the reviewer for careful reviews and valuable comments. Here, we would like to make the following clarifications, explanations, and modifications for the above comments.

(1) This study aims to generate a high-quality global land precipitation dataset. The new products (i.e., MGP-3P and MGP-6P) were compared with other five popular global products (i.e., MSWEP, IMERG-Final, GSMaP-Gauge, ERA5, and CPCU) in global land areas, and were evaluated and compared in mainland China using two high-density ground precipitation observations as the benchmark. The evaluation results show that the MGP-3P product substantially performed better than the other five research-quality products (i.e., MGP-6P, MSWEP, IMERG-Final, GSMaP-Gauge, and ERA5) in terms of most error metrics. This indicates that the MGP-3P

can provide a high-accuracy precipitation dataset for the community and has been

available to the public (https://zenodo.org/record/7386441#.ZBkpxcJBxD-), which

is the main purpose of this work, as well as our contribution to the community.

(2) The novel multi-source precipitation data fusion (MPDF) algorithm uses CC as the

fusion weights for satellite and reanalysis, which is consistent with that of beck et

al., 2017, 2019. It should be pointed out that, however, the MPDF algorithm also

further considers the dependency of satellite and reanalysis precipitation errors on

seasonality, which is conducive to fully taking advantage of the complementary

strengths from satellite and reanalysis data. Furthermore, we used the number of

rain gauges per 0.5° grid cell as the fusion weights for ground precipitation

observations (i.e., CPCU), which can effectively consider the advantage of gauge

observations and avoid precipitation uncertainties because the precipitation of the

grid cells with no gauge observations was not considered in the merged procedure.

More importantly, the spatiotemporal downscaling module used in the MPDF

algorithm is different from that of beck et al., 2017, 2019. Our spatiotemporal

downscaling module not only considered the advantage of satellite and reanalysis

but also took into account the contributions of hourly and 0.25° grid merged

precipitation to corresponding daily and 0.5° grid merged precipitation. The results

of MGP-3P demonstrated this point, although it is not as good as the spatiotemporal

scaling module of IMERG, please see Table 1 (Table 4 in the manuscript).

Admittedly, the MPDF algorithm is very simple, but that also is its advantage that

it is easy for the readers to understand and use it. In fact, the verification results also

demonstrated the MPDF algorithm is effective in considering the advantages from satellite, reanalysis, and gauge data and in improving the quality of precipitation. The MGP-3P substantially performed better than the other five research-quality products (i.e., MGP-6P, MSWEP, IMERG-Final, GSMaP-Gauge, and ERA5) in terms of most error metrics. In particular, the accuracy of the MGP-3P product is obviously better than that of another multi-source data fusion product MSWEP in the evaluation for mainland China. In addition, we found that the quality of the input products is critical for that of the merged precipitation estimates, rather than the number of the input products. This finding can give some valuable information for researchers to customize the multi-source precipitation data fusion algorithms. Given that the MPDF algorithm has the advantages of the above four points and this finding, we believe that this paper gives some new thoughts to the community.

(3) We will carefully revise the writing and organization in the next version.

In summary, this paper is to generate a high-quality global land precipitation product by using the novel MPDF algorithm to provide a high-accuracy precipitation dataset for the community. Currently, the MGP-3P product has been available to the public (https://zenodo.org/record/7386441#.ZBkpxcJBxD-). In addition, we give some new thoughts to the community. The writing issue existing in the previous manuscript might mislead the reviewer. We will significantly improve the quality of the writing in the next version. The authors hope that the above clarifications and explanations can solve the concerns of the reviewer.

**Table 1** Decay rates of five products (i.e., MGP-6P, MGP-3P, IMERG, GSMaP, and ERA5) in terms of POD, FAR, and CC when from a 3 hourly resolution scaling down to an hourly resolution.

| Products | POD (%) | FAR (%) | CC (%) |
|----------|---------|---------|--------|
| MGP-6P | 13.25 | 0.00 | 13.04 |
| MGP-3P | 10.59 | 1.81 | 12.68 |
| IMERG | 7.58 | 4.44 | 10.00 |
| GSMaP | 10.39 | 5.77 | 13.04 |
| ERA5 | 15.00 | 5.00 | 20.00 |

Serious scientific concerns including but are not limited to:

1. what's your basic assumptions for fusing these global dataset? If only consider the CC as the fusion weights, it is too simple and too weak. Beck et al., 2017, 2019 have already investigated such explorations. Please carefully reading such critical references.

Response: we used CC as the fusion weights for satellite and reanalysis, as it can objectively indicate the performance of satellite and reanalysis products. Also, Beck et al. (2017, 2019) verified the effectiveness of this method in merging multiple precipitation estimates. However, the MPDF algorithm did not only consider the CC as the fusion weights and is different from the algorithm of Beck et al. (2017, 2019). The advantages of the MPDF algorithm are reflected in the following aspects:

(1) The MPDF algorithm considers the dependency of precipitation errors on

seasonality because the performance of satellite and reanalysis has a significant

dependency on seasonality. The global land maps of the weights designed to

satellite, reanalysis, and CPCU in four seasons (i.e., MAM, JJA, SON, and DJF)

demonstrated this point, please see Figure 1 (Figure 13 in the manuscript). Thus,

the dependency of satellite and reanalysis precipitation errors on seasonality

considered in the MDFP algorithm is necessary.

[Figure]

**Figure 1** Global land maps of the weights designed to satellite, reanalysis, and CPCU

in four seasons (i.e., MAM, JJA, SON, and DJF) for scheme 2.

(2) The number of rain gauges for each 0.5° grid cell was used as the fusion weights

for ground precipitation observations (i.e., CPCU). This method can fully take

advantage of the strengths of gauge observations as the quality of the ground

precipitation observations depends on the spatial density of the rain gauges

(Krajewski et al., 2000; Villarini and Krajewski, 2007; Roca et al. 2010; Prakash et

al., 2019; Chen et al., 2020). Additionally, this weighting method avoids resulting in precipitation uncertainties caused by gauge observations.

(3) We used the rations between the merged precipitation from different spatiotemporal resolutions as the weights for the spatiotemporal downscaling of precipitation. This method not only considered the advantage of satellite and reanalysis but also took into account the contributions of hourly and 0.25° grid merged precipitation to corresponding daily and 0.5° grid merged precipitation. The results of MGP-3P indicated that this method is effective, although it is not as good as the spatiotemporal scaling module of IMERG, please see Table 1 (Table 4 in the manuscript). This is also one of the key modules of the MPDF algorithm.

(4) The MPDF algorithm is simple but that also is its advantage that it is easy for the readers to understand and use it. More importantly, the MPDF algorithm is effective in improving the accuracy of precipitation, the evaluation results in mainland China demonstrated this point.

The revised manuscript will clarify that Beck et al. (2017, 2019) have used the CC as the fusion weights.

**References**

Krajewski, W. F., Ciach, G. J., McCollum, J. R., and Bacotiu, C.: Initial validation of the Global Precipitation Climatology Project monthly rainfall over the United States. J. Appl. Meteorol. 39(7), 1071-1086, https://doi.org/10.1175/1520-0450(2000)039<1071:IVOTGP>2.0.CO;2, 2000.

Villarini, G., and Krajewski, W. F.: Evaluation of the research version TMPA three-hourly 0.25×0.25 rainfall estimates over Oklahoma. Geophys. Res. Lett. 34, L05402, https://doi.org/10.1029/2006GL029147, 2007.

Roca, R., Chambon, P., Jobard, I., Kirstetter, P. E., Gosset, M., and Bergès, J. C.: Comparing satellite and surface rainfall products over West Africa at meteorologically relevant scales during the AMMA campaign using error estimates. J. Appl. Meteorol. Clim. 49(4), 715-731, https://doi.org/10.1175/2009JAMC2318.1, 2010.

Prakash, S., Seshadri, A., Srinivasan, J., and Pai, D. S.: A new parameter to assess impact of rain gauge density on uncertainty in the estimate of monthly rainfall over India. J. Hydrometeorol. 20(5), 821-832, https://doi.org/10.1175/JHM-D-18-0161.1, 2019.

Chen, H., Yong, B., Qi, W., Wu, H., Ren, L., and Hong, Y.: Investigating the evaluation uncertainty for satellite precipitation estimates based on two different ground precipitation observation products. J. Hydrometeorol. 21(11), 2595-2606, https://doi.org/10.1175/JHM-D-20-0103.1, 2020.

2. The title tells that this study aims to public a global dataset, however, it only provides precipitation estimates over Land. It is really not rigorous. Please take care of such issues.

Response: thank you. We will modify this issue in the next version. The title will be revised to "MGP: a new 1-hourly 0.25° global land precipitation product (2000-2020)

based on multi-source precipitation data fusion".

3. How did the authors consider the negative effects from the different input precipitation estimates, especially in terms of the precipitation events? The weights based on CC could be only achieved at daily scale. So how did you consider the systematic and random errors at hourly scales from the input datasets?

Response: those are good questions. In fact, the MPDF algorithm only considers the precipitation amount, without considering precipitation events. Notably, the MPDF algorithm fully takes advantage of the complementary strengths from satellite, reanalysis, and gauge data, but also propagates negative effects from different input products into merged precipitation estimates. For instance, the FAR of the MGP products is higher than that of most global precipitation products, which is due primarily to the false precipitation of ERA5 being propagated into the MGP suite. In addition, the MGP suite is not the best in terms of the total bias, as the rainfall amount of the satellite and ERA5 reanalysis was not corrected before the weighted merging. Those issues of the MPDF algorithm were found, and we have discussed those in the manuscript, please see lines 610-622.

The weights at hourly scale are from those at daily scale, because the weights based on CC could be only achieved at daily scale. The relative size of ratio for CC between different input products could be well consistent across different time scales, as the

quality of all satellite precipitation products decreases when scaled down to shorter time periods (Tan et al., 2017). However, we believe that the differences would be found in areas with complex terrain and strong spatiotemporal heterogeneity of precipitation. We will discuss this in the revised manuscript.

**Reference**

Tan, J., Petersen, W.A., Kirstetter, P.E., and Tian, Y.: Performance of IMERG as a function of spatiotemporal scale. J. Hydrometeorol. 18 (2), 307-319, https://doi.org/10.1175/JHM-D-16-0174.1, 2017.

4. The resolutions of the MPG is very strange with 1-hourly and 0.25°. most popular satellite and reanalysis precipitation datasets have finer resolutions at 1-hourly and 0.1°, for instance, IMERG, GSMaP, and ERA5-Land. Particularly, PERSIANN-CCS is quiet finer with resolutions half-hourly and 0.04°. So what's your purpose of the resolutions at 1-hourly and 0.25°?

Response: thank you for the comments. The spatiotemporal resolution is mainly determined by the accuracy of the MGP-3P product. We noted that the spatiotemporal downscaling module can maintain the accuracy of the MGP in a favorable position at 0.25° spatial and hourly temporal resolutions. In addition, if the spatial resolution of the MGP-3P product is finer (< 0.25°), only IMERG-Late can be used as input for the spatiotemporal downscaling module, which cannot ensure the accuracy of the MGP-3P

product at a finer spatial resolution. Consequently, the purpose of the resolutions at 1-hourly and 0.25° is to ensure that the accuracy of the MGP-3P product.

5. The authors seems to have not enough background information on such satellite-based and reanalysis-based precipitation datasets. For instance, the most important aim of PERSIANN-CCS is to capture the first glimpse of the possible precipitation, not the quality. The authors considered the PERSIANN-CCS to provide what information at 1-hourly and 0.25° for developing the qualified research level precipitation product?

Response: thank you for the comments. In this study, we would like to answer the question of which is crucial for the quality of merged precipitation estimates: the number of input products or the quality of input products? Therefore, we have designed six input sources for merging scheme 1 to generate MGP-6P, without considering the quality of input products, whereas we consider the quality of input products in merging scheme 2 for generating a research level precipitation product MGP-3P, merging the best satellite-only IMERG-Late and model-based ERA5 reanalysis precipitation products. A comparison between MGP-3P and MGP-6P was used to answer the above question. Finally, we found that the quality of input products is critical for that of merged precipitation estimates, rather than the number of input products. The detailed information can be seen in section 4 and lines 302-328.

6. In terms of evaluation and comparison, the results have various weak aspects and do not make me convinced, especially due to the black box merging model: (1) only evaluated at mainland China? Would it be reasonable to represent the global situations? (2) what are the reasons for improving the POD and FAR of MGP-6P and MGP-3P? just because there were merged based on CC only achieved at daily scales? and (3) why not evaluate and compare these precipitation products over CONUS where have enough ground observations for public?

Response: thank you for the comments. We only performed the evaluation and comparison in mainland China, due to lacking the high-quality ground precipitation observations for the rest of the world. We admit that the evaluation results from mainland China cannot represent the global situation, and we also indicated the limitations of evaluation in the manuscript, please see lines 631-649. We are very grateful to the reviewers for the suggestions that evaluate and compare the precipitation products over CONUS. We accept this suggestion and will do it in the revised manuscript.

In terms of the reasons for improving the POD and FAR of MGP-6P and MGP-3P, we believe the reasons mainly include the following aspects:

(1) The MPDF algorithm fully take advantage of the strengths from ERA5 which has the best detection capability compared to satellite-only precipitation products (Jiang et al., 2021; Chen et al., 2023). We believe that this is the main reason for improving

the POD of the MGP suite. This has been discussed in the manuscript, please see lines 425-426.

(2) The weighting method considered the dependency of satellite and reanalysis precipitation errors on seasonality, which fully takes advantage of the complementary strengths from satellite and reanalysis data.

(3) The weighting method of gauge observations (i.e., using the number of rain gauges as the fusion weights) is effective in considering the advantage of gauge observations, and the precipitation estimates in the grid cells with no rain gauges were not merged, avoiding precipitation uncertainties caused by gauge observations.

(4) The spatiotemporal downscaling module not only considered the advantage of satellite and reanalysis data but also took into account the contributions of hourly and 0.25° grid merged precipitation to corresponding daily and 0.5° grid merged precipitation, which improves the performance of the MGP in terms of POD.

**Reference**

Jiang, Q., Li, W., Fan, Z., He, X., Sun, W., Chen, S., Wen, J., Gao, J., and Wang, J.: Evaluation of the ERA5 reanalysis precipitation dataset over Chinese Mainland. J. Hydro. 595, 125660, https://doi.org/10.1016/j.jhydrol.2020.125660, 2021.

Chen, H., Wen, D., Du, Y., Xiong, L., and Wang, L.: Errors of five satellite precipitation products for different rainfall intensities. Atmos. Res. 285, 106622, https://doi.org/10.1016/j.atmosres.2023.106622, 2023.

---

## Author Comment (AC2)

**Response for Reviewer #2**

High-accurate precipitation data are essential for various aspects and estimating precipitation is challenging especially in complex-terrain and ungauged regions. This work produced a global precipitation dataset (0.25°, 1 hourly) by weighted average of multiple precipitation datasets. However, great improvements are needed in terms of the merging method and validation of the dataset. The main concerns are as follows:

Response: We would like to thank you for your constructive comments on our manuscript. Your insightful review has improved our manuscript considerably. Below is a point-by-point response to your comments.

1) Validation of the produced dataset is inadequate. The authors produced a global precipitation dataset. However, the validation with gauge observations was only conducted in Chinese Mainland, which does not support its accuracy in other regions of the world. I suggest the authors perform a more convincing validation using worldwide observations.

Response: Thank you for your constructive comments and insightful suggestions. We agree with the reviewer that the validation of the MGP suite should be performed on a global scale. However, an independent high-quality global ground precipitation dataset is lacking, which is a problem faced by global scholars, not limited to us. Taking an example, the MSWEP product was only validated in the United States (please see Beck

et al., 2019), due to lacking high-quality independent global ground precipitation observations for the public. We really want to validate the MGP products on a global scale, which is crucial for the product application and further improvement of the quality. In the revised manuscript, we will add validation in the United States, because we are getting an independent high-quality ground precipitation dataset here. Meanwhile, we are looking for high-quality independent ground precipitation datasets from around the world. If we can get an independent high-quality ground precipitation dataset in other areas, we will add more validation in the revised manuscript.

2) Why CC was used as the weight for merging various datasets rather than other metrics like root mean square error (RMSE), Kling–Gupta efficiency (KGE), Nash-Sutcliffe efficiency (NSE), et al. In the case where a dataset is highly consistent with the observations in terms of the temporal trend but has a systematic error, this method may lead to a systematic bias in the merging output.

Response: Thank you for your comments. The correlation coefficient (CC) can accurately quantify the performance of evaluated global products, and its range of values is from 0 to 1. Thus, CC is very suitable for determining the weights of global precipitation products. Additionally, Beck et al. (2019) also verified the effectiveness of CC in determining the fusion weights. We agree with the reviewer that the method using CC to determine fusion weights may lead to a systematic bias in the merging output. We will discuss this a little bit in the next version.

We are very grateful to the reviewer for the insightful suggestions. RMSE is related to precipitation intensity. Large differences in space and time exist between satellite-based and model-based precipitation intensity. The method using RMSE to determine fusion weights can consider the dependency of precipitation errors on precipitation intensity but also might introduce an intensity-based bias in the merging output. Considering the respective strengths of CC and RMSE, we accepted the suggestion and add using RMSE to determine fusion weights in the revised manuscript for comparing the performance of these two methods in determining fusion weights. KGE and NSE might be potential weighting methods, but their ranges are negative infinity to 1. The range is too large, and it is also difficult to get a reasonable fusion weight when KGE or NSE is a negative number. These limitations are unfavorable for accurately identifying the error characteristics in each dataset. We will provide a detailed explanation in the revised manuscript of why we used CC and RMSE for weighting and discuss possible limitations with CC and RMSE in weighting based on verification results.

3) The authors repeated the merging procedures three times at different spatial and temporal scales and then spatially and temporally downscale the coarse dataset. What is the underlying logic for applying such a strategy?

Response: Thank you for the comments. The results of Tan et al. (2017) indicated that compared with spatial downscaling, the loss in performance of precipitation products is large in temporal downscaling. To ensure the high accuracy of the produced product,

consequently, we first conducted spatial downscaling and then conducted temporal downscaling, for generating high-quality MGP products. We have added the explanations; the modification is represented as follows: Tan et al. (2017) indicated that the reduction of accuracy for precipitation products in temporal downscaling is greater than that in spatial downscaling. To obtain high-accuracy global merged precipitation estimates, therefore, we first conducted spatial downscaling to minimize the loss of the quality of precipitation caused by spatiotemporal downscaling.

4) The authors emphasized that considering the seasonality of errors in the merging procedures is novel. However, no evidence was provided in the manuscript to support the novelty or added value of considering the seasonality of errors in precipitation datasets.

Response: Thank you for pointing this out. Many evaluation studies found that the errors in satellite and reanalysis precipitation products have a significant dependency on seasonality (e.g., Tian et al., 2009; Jiang et al., 2021; Chen et al., 2021). In addition, Global land maps of the weights for satellite and reanalysis data have obvious differences (please see Figure.1). For instance, ERA5 in DJF and MAM seasons has large weights in high latitude and cold mountainous areas, whereas the weights of satellite (i.e., IMERG-Late) in DJF season are lower (<30%) in high latitude areas. Therefore, it is necessary to consider the dependency of precipitation errors on seasonality in the merging algorithms, which can accurately identify the error of

seasonality in each dataset. The added explanations (modifications) were presented as follows: (1) Many evaluation studies found that the errors in satellite and reanalysis precipitation products have a significant dependency on seasonality (e.g., Tian et al., 2009; Jiang et al., 2021; Chen et al., 2021). Therefore, the precipitation amounts of satellite and reanalysis precipitation products were separated into four seasons (i.e., MAM (March-May), JJA (June-August), SON (September-November), and DJF (December-February)); (2) In addition, Global land maps of the weights for satellite and reanalysis data have obvious differences. For instance, ERA5 in DJF and MAM seasons has large weights in high latitude and cold mountainous areas, whereas the weights of satellite (i.e., IMERG-Late) in DJF season are lower (<30%) in high latitude areas.

[Figure]

Figure 1. Global land maps of the weights designed to satellite, reanalysis, and CPCU in four seasons (i.e., MAM, JJA, SON, and DJF).

References

Tian, Y., Peters-Lidard, C.D., Eylander, J.B., Joyce, R.J., Huffman, G.J., Adler, R.F., Hsu, K.L., Turk, F.J., Garcia, M., and Zeng, J.: Component analysis of errors in satellitebased precipitation estimates. J. Geophys. Res.-Atmos. 114 (D24), https://doi.org/10.1029/2009JD011949, 2009.

Jiang, Q., Li, W., Fan, Z., He, X., Sun, W., Chen, S., Wen, J., Gao, J., and Wang, J.: Evaluation of the ERA5 reanalysis precipitation dataset over Chinese Mainland. J. Hydro. 595, 125660, https://doi.org/10.1016/j.jhydrol.2020.125660, 2021.

Chen, H., Yong, B., Kirstetter, P. E., Wang, L., and Hong, Y.: Global component analysis of errors in three satellite-only global precipitation estimates. Hydrol. Earth Syst. Sci. 25(6), 3087-3104, https://doi.org/10.5194/hess-25-3087-2021, 2021.

5) The comparison between MGP-6P and MGP-3P makes no sense. If another group of datasets was selected for the merging, the results may differ evidently from those in the manuscript. The key to this problem is not the number of datasets that are applied, but whether the merging method can accurately identify the error characteristics in each dataset.

Response: Many thanks for your comments. The comparison between MGP-3P and MGP-6P was used to answer the question of which is crucial for the quality of merged precipitation estimates: the number of input products or the quality of input products? Therefore, we designed six input sources for merging scheme 1 to generate MGP-6P,

without considering the quality of input products, whereas we considered the quality of input products in merging scheme 2 for generating a research level precipitation product MGP-3P, merging the satellite-only IMERG-Late and ERA5 reanalysis precipitation products. We agree with the reviewer that it is important that the merging method can accurately identify the error characteristics in each dataset. However, most algorithms often combine the advantage of the strengths of different data with their shortcomings into the merging output. At this point, the quality of the input products is also very important to that of merging output. That is why we performed the comparison between MGP-3P and MGP-6P. In the revised version, we will delete merging scheme 1 and keep merging scheme 2, and hence the comparison between MGP-6P and MGP-3P will be removed.

6) Given that a large number of global datasets have been available with a spatial resolution of 0.1° or finer, the produced dataset with a spatial resolution of 0.25°may not be competitive in the science community.

Response: Thank you for your comments. In version 1 of MGP, the accuracy of MGP is the primary consideration. Considering the finest spatial resolution of ERA5 is 0.25°, the purpose of the spatial resolution at 0.25° is to ensure the accuracy of the MGP-3P product. If the spatial resolution of MGP is 0.1°, only IMERG-Late can be used as input in the spatial downscaling when the spatial resolution is scaled down from 0.25°to 0.1°, which could not guarantee the accuracy of MGP-3P. We agree with the reviewer that

on the premise of ensuring the accuracy of MGP, the finer the resolution of MGP, the more competitive it becomes. In the revised manuscript, we will try to use ERA5-Land (0.1°) instead of ERA5 (0.25°) as one of the input products for the MPDF algorithm for generating MGP with a finer spatial resolution (0.1°, hourly).

**Some specific comments:**

7) Line 74-75, "the quality of gauge observations is extremely dependent on the spatial density of the rain gauges": the quality of gridded data interpolated from gauge observations, rather than the gauge observation itself, depends on gauge density.

Response: Thanks. We have revised this sentence as follows: "Nevertheless, the quality of gridded data interpolated from gauge observations is extremely dependent on the spatial density of the rain gauges.".

8) Line 143-145, "spatial interpolation was proved to be an effective method in improving the quality of global satellite and reanalysis precipitation estimates": There is no definitive relationship between spatial interpolation and improving the quality of precipitation datasets.

Response: Thank you for pointing this out. We have removed this incorrect description.

9) There are many confusing sentences in the manuscript, e.g. Line 177-182, Line 242-243, Line 273-275, Line 467-469, Line 526-527.

Response: Thank you for pointing those confusing sentences out. Line 177-182 was revised as follows: "The quality of the MSWEP product mainly depends on that of the ground-based precipitation observations as the weights designed for ground-based precipitation observations are significantly larger than those designed for satellite- and reanalysis-based precipitation estimates."; Line 242-243 was revised as follows: "The spatial distribution of rain gauges used in the CPCU system can be seen in Fig. 2a."; Line 273-275 was revised as follows: "The precipitation estimates at a grid cell come from CPCU precipitation observations when rain gauges captured precipitation occurrences at the same grid cell but satellite and reanalysis did not."; Line 467-469 was revised as follows: "the MSWEP estimates were derived from high-quality ground-based precipitation observations by using the spatiotemporal downscaling technique (Beck et al., 2017, 2019)."; Line 526-527 was revised as follows: Notably, false precipitation was not observed from evaluated global precipitation products as the rainfall intensities were from the ground references.

We will carefully check the manuscript to eliminate all issues.

10) The used data are not fully introduced, especially for the CPCU dataset.

Response: Thank you for pointing this issue out. We have added the introduction of the used data. The modifications were presented as follows: Three different global precipitation products, i.e., satellite-only IMERG-Late (0.1°, 0.5 hourly; Huffman et al., 2019), ERA5-Land reanalysis (0.1°, hourly; C3S, 2017; Hersbach et al., 2020), and ground-based Climate Precipitation Center unified (CPCU; 0.5°, daily; Xie et al., 2007; Chen et al., 2008), were used as inputs for the new algorithm. IMERG-Late is a multi-satellite merged precipitation product that blends microwave and infrared data. ERA5-Land is a model-based precipitation dataset, with finer spatiotemporal resolutions (0.1°, hourly) compared with other model-based precipitation products. In terms of CPCU, it was generated using optimal interpolation based on > 17000 gauges, and its precipitation estimates cover quasi-global (50°S-50°N) land areas (Xie et al., 2007; Chen et al., 2008). The reasons for the selection of these precipitation products are that IMERG-Late performed better than other satellite-only precipitation products in most areas of the world (Chen et al., 2020b, 2021), whereas ERA5-Land has the finest spatiotemporal resolution compared with other model-based reanalysis precipitation products.

11) Line 412-414: it is not easy to see the differences from Fig. 3. It is suggested to present the differences between these global datasets and the produced dataset.

Response: Thank you for your insightful suggestion. We accepted your suggestion, and the maps of differences between these global datasets and the produced dataset will be

provided in the revised manuscript. Notably, the analysis based on new results will be

provided.

12) Section 4.2. Comparing the quality of precipitation datasets on different temporal

scales makes no sense, because it is well acknowledged that precipitation product

performs worse on a shorter temporal scale than on a longer scale.

Response: Thank you for the comments. We agree with the reviewer. We have deleted

Section 4.2.